# Detect What You Need: Chain-of-Causal Reasoning for 3D Intent Grounding

## Abstract

Accurately matching human intentions in 3D space is an important goal of artificial intelligence. Recently, 3D Intension Grounding (3D-IG) is proposed, aiming to localize target 3D objects that match the given natural language intent. Compared with traditional visual grounding with a clear goal, the intent is abstract and difficult to understand, posing enormous challenges in detecting corresponding 3D objects. To this end, towards this task, the model is required to infer the functional attributes of objects from the captured non-descriptive intent and then precisely align attributes to object features. During this process, existing methods rely on implicit matching, which often suffers from logical gaps. As a result, they fail to establish a clear and interpretable causal reasoning between intention and object, ultimately lowering the robustness and generalizability of the model. To tackle these challenges, we propose a new method, i.e., Chain-of-Causal Reasoning, which performs intent parsing and grounding along the causal chain. Specifically, the method decomposes complex intentions step by step along the causal chain into functional requirements, explicitly prioritizing and clarifying latent needs, thereby forming a causal chain from abstract intentions to object attributes and enhancing the accuracy of intent understanding. Based on this causal chain, we construct an explicit causal graph to establish clear logical relationships between functional requirements and object attributes. Finally, a causal–visual feature alignment mechanism is introduced, which aligns causal features with the geometric–semantic features of 3D point clouds, enabling bidirectional verification between semantic reasoning and visual evidence. Extensive experiments in 3D Intention Grounding and 3D Visual Grounding tasks demonstrate that our method effectively enhances intent understanding and improves object localization.

## 1 Introduction

In real-world human-computer interaction scenarios, particularly in applications involving embodied intelligence Lin et al. (2025), humans typically express their needs through natural language (e.g., 'I need back support'), rather than providing explicit object references Shi et al. (2024b); Guo et al. (2025) (e.g., 'the pillow on the sofa'). Therefore, compared to the traditional 3D Visual Grounding task Shi et al. (2024b); Guo et al. (2025), the recently proposed 3D Intent Grounding (3D-IG) task Kang et al. (2024) is more suitable for real-world applications. The core objective of this task is to enable models to directly locate target objects that meet functional requirements based on natural language intent input, and accurately perform localization and matching within 3D scenes.

The core challenge of the 3D-IG task lies in the abstract nature of human intent. Unlike the traditional Visual Grounding task, which relies on explicit object categories or visual descriptions, the intent texts processed by 3D-IG do not point to specific object characteristics. Instead, they focus on functional attributes and usage contexts. This requires the model first to infer implicit functional requirements from vague expressions and then match them with 3D object features, significantly increasing the complexity of the task. Existing methods Kang et al. (2024) match target objects by establishing semantic correspondences between 'verbs and objects,' relying on direct alignment between verbs and objects, as shown on the left side of Figure 1. However, this direct semantic association introduces a clear logical gap, as the model does not understand why a particular object should match a given intent. Specifically, the model can only match at the semantic level but fails to infer whether an object truly satisfies the implicit functional requirements of the intent. For example, when faced with the intent 'I need something to support my back,' the model may match 'support' and 'back' with multiple objects, such as a sofa, pillow, or chair, but it cannot determine which object actually

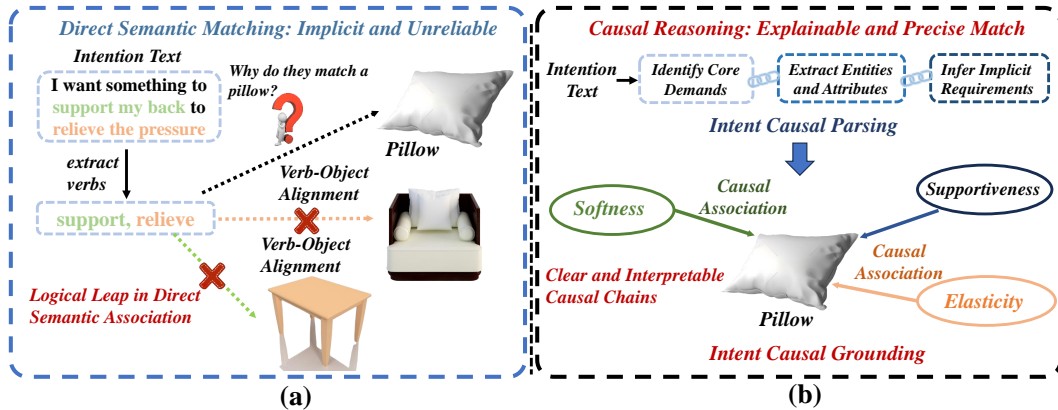

Figure 1: (a) The above illustrates the limitations of existing methods, where direct association through semantic matching leads to significant logical leaps. Below, our approach achieves step-by-step causal reasoning through stepwise parsing. (b) The detailed workflow of our Chain-of-Causal Reasoning method constructs multiple causal associations between intentions and objects through Intent Causal Parsing and Intent Causal Grounding, enabling object reasoning and localization.

provides 'back support,' leading to potential mismatches and incorrect localization in open-world scenarios. Therefore, existing methods often lack a deep understanding of the causal relationships between intent and objects, limiting the robustness and generalization capability of the model.

To address these issues, we propose a new method, i.e., Chain-of-Causal Reasoning, which performs intent parsing and grounding along the causal chain. Specifically, our method first employs Intent Causal Parsing to decompose natural language intentions step by step along the causal chain into functional requirements, as illustrated in the upper-right part of Fig. 1. By explicitly defining and prioritizing latent needs, the module enables the model to more accurately capture the user's intent. For example, given the intent of natural language, 'I want something to support my back to relieve pressure', the model first identifies the core requirements: 'support my back' and 'relieve pressure'. It then further infers the corresponding physical properties: the former requires the object to provide adequate support, while the latter implies that the object should be soft. Through this stepwise, Chain-of-Causal reasoning mechanism, the model transforms complex intentions into explicit functional requirements, forming a causal chain from abstract intent to object attributes. This progressive reasoning along the causal chain allows the model to consistently understand complex intentions, avoiding misunderstandings caused by semantic ambiguity. Next, based on the causal chain, the parsed functional requirements are represented as intermediate nodes in a causal graph, establishing explicit links from natural language intent to object selection. For example, the intent 'support my back' is decomposed into functional nodes such as 'supportiveness' and 'softness', and causal mappings are established between these functional nodes and candidate objects, as shown in the lower right part of Fig. 1. This design not only avoids the logical gaps present in traditional semantic-matching methods but also provides interpretable causal reasoning for object detection, enhancing the model's robustness and explainability in open-world scenarios. Finally, a Causal–Visual Feature Alignment is introduced, encoding functional requirements as causal features and aligning them with 3D point cloud features to enable bidirectional verification between causal reasoning and visual evidence. Extensive experiments in 3D Intention Grounding and 3D Visual Grounding demonstrate that our method effectively enhances intent understanding and improves object localization.

To summarize, our contributions are as follows:

• We propose a new method, Chain-of-Causal Reasoning, which progressively decomposes complex intentions along the causal chain and establishes explicit mappings from intent to object attributes in a causal graph, enabling interpretable and robust 3D object localization.

• We design a causal–visual feature alignment that encodes functional requirements as causal features and aligns them with 3D point cloud features, enabling bidirectional verification between causal reasoning and visual evidence.

• We validated the effectiveness of our method on 3D Intention Grounding and 3D Visual Grounding, achieving state-of-the-art performance.

## 2 RELATED WORKS

### 2.1 3D VISUAL GROUNDING

3D Visual Grounding Shi et al. (2024b); Qian et al. (2024a) aims to locate target objects within a 3D scene based on natural language descriptions. Existing methods are typically categorized into two-stage and single-stage approaches, with the core goal of aligning visual and textual features to identify the target object. For instance, the PQ3D Zhu et al. (2024) method enhances a unified, prompt-driven framework that connects various 3D representations, enabling versatile 3D vision-language understanding from perception to reasoning. The TSP3D Guo et al. (2025) method addresses the inefficiency of traditional approaches and the difficulty of interaction between 3D scene and text features through text-guided pruning and completion mechanisms. While these methods achieve good performance in visual localization tasks, they typically require providing the object's category name for direct localization. However, in real-world applications, humans generally express intent through natural language, rather than providing explicit object references. Therefore, compared to 3D Visual Grounding, the 3D Intention Grounding task is more aligned with practical application scenarios, as it can infer and localize target objects based on functional requirements directly from natural language. Further discussion on Causal Learning is provided in Appendix B.1.1.

### 2.2 3D INTENTION GROUNDING

3D Intention Grounding is a task aimed at understanding and locating objects in a 3D scene based on human intent expressed through natural language. Unlike traditional visual localization methods, which rely on direct references to specific objects or their categories, 3D Intention Grounding focuses on understanding the functionality and properties that users expect in objects, such as 'support for my back' or 'softness,' expressed through free-text descriptions. The existing IntentNet Kang et al. (2024) method uses verb-object alignment to locate objects within the intent, relying directly on the alignment between actions and targets. However, this approach does not infer from the intent text directly through the model; instead, it depends on prior knowledge of object functions in the 3D scene and dataset annotations, leading to logical leaps in semantics. Due to the lack of explicit modeling of common-sense logic, existing methods tend to fail in open-world scenarios, particularly when encountering unannotated objects. The model lacks causal logic support, making it difficult to determine whether an object satisfies the intent, resulting in decreased detection accuracy. To address this issue, we propose a method, Chain-of-Causal Reasoning, which progressively decomposes complex intentions along the causal chain and establishes explicit mappings from intents to object attributes in a causal graph, enabling interpretable and robust 3D object localization.

### 2.3 CAUSAL GRAPH MODELING

In machine learning, incorporating causality enhances both the learnability and interpretability of models. Causal Graph Modeling has been widely applied to 2D tasks such as image classification Hu et al. (2025), object detection Li et al. (2025a), explicitly modeling causal relationships among attributes, relations, or confounders to improve interpretability and generalization. However, in 3D vision, causal modeling remains challenging due to the sparsity and irregularity of point cloud data. In 3D point cloud classification, CausalPC Huang et al. (2024c) constructs a structural causal model to purify inputs and maintain classifier performance. For 3D Novel Class Discovery, JLCR Li et al. (2025b) removes confounders via causal representation prototypes and leverages graph-based causal reasoning to enable knowledge transfer and precise segmentation from base to novel classes. In 3D visual grounding, MA2TransVG Xu et al. (2024) uses an Attribute Causal Analysis Module to quantify the causal effects of multiple attributes on predictions, enhancing multimodal feature alignment and fine-grained localization. However, this attribute-based causal matching relies on explicit visual cues in the text and is not applicable to abstract intent understanding. In contrast, our method progressively decomposes complex intentions along a causal chain, explicitly mapping textual intents to visual attributes, enabling more robust 3D intent understanding.

## 3 METHOD

The overall framework of our Chain-of-Causal Reasoning (CoCR) method is illustrated in Fig. 2. It consists of three key modules: (i) Intent Causal Parsing, which decomposes complex intentions step by step along the Chain-of-Causal; (ii) Intent Causal Grounding, which establishes explicit causal links to capture latent dependencies between functional requirements and candidate objects; and (iii) Causal–Visual Feature Alignment, which iteratively updates the causal graph and enables robust cross-modal reasoning.

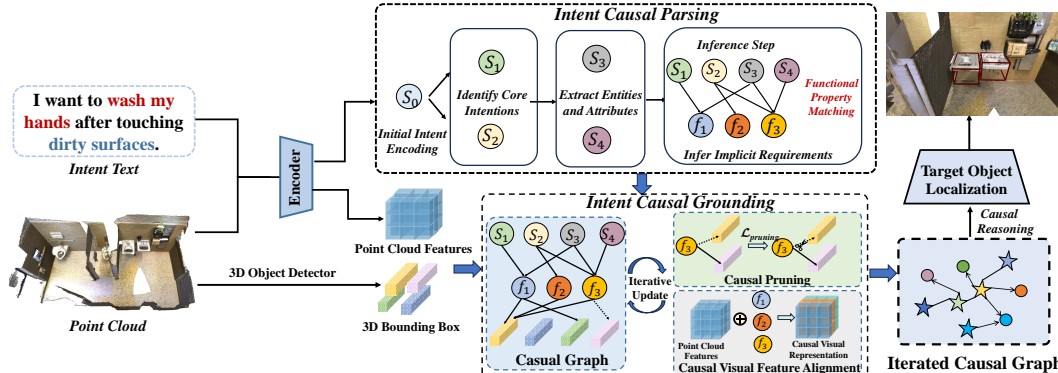

Figure 2: **The overall architecture**. First, point cloud data and intent text are encoded by the model. The intent text is processed by the Intent Causal Parsing module for step-wise decomposition, transforming the natural language intent into functional requirements. Subsequently, Intent Causal Grounding is performed through the construction of a causal graph, defining the causal relationships between functional requirements and object attributes. The graph is iteratively updated via causal–visual feature alignment and causal pruning. Finally, causal reasoning is conducted based on the updated graph to achieve object selection and localization.

## 3.1 INTENT CAUSAL PARSING

To analyze complex natural language intentions and extract actionable functional requirements, we design the Intent Causal Parsing, which performs step-wise decomposition of complex intentions along the Chain-of-Causal. The module breaks down intentions into hierarchical functional requirements, gradually uncovering implicit information, and explicitly prioritizes and clarifies latent needs to help the model better understand human intentions. Given a natural language intent $Q$, such as 'I want something that can support my back to relieve pressure', the model first uses the fine-tuned T5-small model Raffel et al. (2020) with prompt templates to parse and decompose the intent, with more details provided in Appendix B.2, generating a step-by-step reasoning sequence $\mathcal{S}$, as follows:

$$\mathcal{S} = \{s_1, s_2, \ldots, s_t\}, \tag{1}$$

where each $s_t$ represents an intermediate reasoning step in the process, for example, from the intent 'I want something to support my back to relieve the pressure,' the model first identifies the core needs: 'support my back' and 'relieve the pressure.' Next, the model further infers specific physical properties based on these core requirements. For instance, from 'support my back,' the model infers that the object needs to have supportive properties, while from 'relieve the pressure,' it concludes that the object should have softness to alleviate pressure. Eventually, the model synthesizes these into a complete set of functional requirements for the object, including both supportiveness and softness, along with the corresponding physical characteristics. To align each reasoning step with its corresponding functional requirement, we encode each reasoning step $s_t$ as a vector $h_i \in \mathbb{R}^{d_s}$, where $d_s$ is the dimensionality of the encoded vector, set to 512. The model then uses cosine similarity to measure the alignment between each reasoning step and all functional requirement nodes $f_j$. The cosine similarity is computed as follows:

$$\cos(h_i, f_j) = \frac{h_i \cdot f_j}{\|h_i\|\|f_j\|}, \tag{2}$$

where $\cdot$ represents the vector dot product, and $\|\cdot\|$ denotes the L2 norm of the vector. By calculating the cosine similarity between each reasoning step $s_t$ and all functional requirement nodes $f_j$, the model determines the optimal match for each reasoning step with a functional requirement node. For example, the reasoning step 'support my back' would match the functional requirement node 'Supportiveness,' and the reasoning step 'relieve the pressure' would match the functional requirement node 'Softness' and 'Elasticity'. To further transform these similarities into a normalized probability distribution over functional requirements, the model applies a softmax function:

$$P(f_j \mid Q) = \frac{\exp(\cos(h_i, f_j))}{\sum_{k=1}^{K} \exp(\cos(h_i, f_k))}, \tag{3}$$

---

**Algorithm 1:** Chain-of-Causal Reasoning for 3D Intent Grounding

---

**Input:** 3D point cloud $V$, natural language intent $Q$, object category set $\mathcal{O} = \{o_1, \ldots, o_m\}$
**Output:** Matched object $o^*$

1 **Step 1: Intent Causal Parsing**
2 Use pre-trained T5-small model to generate a stepwise reasoning sequence $\mathcal{S} = \{s_1, \ldots, s_t\}$
   from $Q$;
3 Infer functional requirement set $F = \{f_1, \ldots, f_K\}$ from $\mathcal{S}$;
4 Compute probability distribution $P(f_j|Q)$ using cosine similarity between $s_t$ and $f_j$;
5 **Step 2: Intent Causal Grounding**
6 Construct a causal graph $\mathcal{G} = (\mathcal{V}, \mathcal{E})$ with nodes $\mathcal{V} = \{Q, F, \mathcal{O}\}$ and edges $\mathcal{E}$;
7 Model causal dependency between $f_j$ and $o_m$ as $P(o_m|f_j)$;
8 Prune the causal graph using a loss function $\mathcal{L}_{pruning}(\theta)$ to remove edges with causal weight
   $w_{ij} < \theta$;
9 Calculate preliminary matching score $g_m = \sum_{j=1}^{K} P(f_j|Q) \cdot \cos(o_m, f_j)$;
10 **Step 3: Causal-Visual Feature Alignment**
11 Extract scene representation $V \in \mathbb{R}^{m \times d}$ using a 3D point cloud encoder;
12 Obtain causal features $C \in \mathbb{R}^{K \times d}$ where $C_j = P(f_j \mid Q) \cdot \text{Embed}(f_j)$;
13 Align and fuse features using cross-attention: $\tilde{V} = \text{CrossAttn}(V, C)$;
14 Introduce bidirectional consistency constraint $\mathcal{L}_{\text{cons}}$ to enforce mutual verification;
15 **Step 4: Final Object Selection**
16 Compute final matching score for each candidate object $o_m$ based on fused features $\tilde{V}$ and causal
   features $C$;
17    $g_m = \text{Match}(\tilde{v}_m, C), \quad m = 1, \ldots, m$;
18 Select the object with the highest score as the final output: $o^* = \arg\max_m g_m$;
19 **return** $o^*$

---

where $K$ denotes the total number of functional requirement nodes, which gradually increases during training. The probability distribution is obtained by applying a softmax over the cosine similarities between the encoded intent feature $h_i$ and all functional requirement embeddings $f_1, \ldots, f_K$, representing the likelihood of each functional requirement given the intent. The Intent Causal Parsing module thus outputs $P(f_j \mid Q)$, which is directly used in the subsequent Intent Causal Grounding stage to establish causal links between functional requirements and candidate objects.

### 3.2 INTENT CAUSAL GROUNDING

After obtaining the probability distribution $P(f_j \mid Q)$ from Intent Causal Parsing, we construct a causal graph based on the correspondence between intents and functional requirements to perform Intent Causal Grounding. Explicitly modeling the causal relationships among intents, functional nodes, and candidate objects enables the model to accurately map functional requirements to object features, avoiding gaps from relying solely on semantic matching. Specifically, we map the functional requirement set $F = \{f_1, f_2, \ldots, f_K\}$ to intermediate nodes in the causal graph and the object category set $\mathcal{O} = \{o_1, o_2, \ldots, o_m\}$ to candidate terminal nodes. The causal graph $\mathcal{G} = (\mathcal{V}, \mathcal{E})$ consists of a node set $\mathcal{V} = \{Q, F, \mathcal{O}\}$ and an edge set $\mathcal{E}$, where edges represent 'causal dependency' relationships. The intent node $Q$ is mapped to functional requirement nodes $f_j$, and further, functional requirement nodes are mapped to candidate object nodes $o_m$. The mapping relationship is represented by the following conditional probability:

$$P(o_m \mid Q) = \sum_{j=1}^{K} P(o_m \mid f_j) \cdot P(f_j \mid Q), \tag{4}$$

where $P(f_j \mid Q)$ is the probability of parsing the functional requirement from the intent $Q$, and $P(o_m \mid f_j)$ represents the causal dependency strength between the functional requirements $f_j$ and the candidate objects $o_m$.

To enhance the efficiency of the causal graph and remove redundant causal relationships, we introduce a causal pruning constraint Dong et al. (2024), removing edges in the causal graph that are irrelevant

to the functional requirements. Specifically, we set a causal pruning threshold $\theta$ and remove edges where the causal weight $w_{ij}$ is below this threshold, reducing unnecessary dependencies. This process is implemented by the following loss function:

$$\mathcal{L}_{pruning}(\theta) = \sum_{(c_i, n_j) \in \mathcal{E}} \mathbb{I}(w_{ij} < \theta) \cdot w_{ij}^2, \tag{5}$$

where $\mathbb{I}(w_{ij} < \theta)$ is an indicator function that removes the edge when $w_{ij} < \theta$, and $w_{ij}^2$ penalizes weak connections. Through this causal pruning constraint, the model automatically removes irrelevant object properties and redundant relationships, improving the accuracy of the causal graph and avoiding interference from unrelated attributes during inference. We verified experimentally that the best results are achieved when $\theta = 0.2$.

Finally, using the optimized causal graph $\mathcal{G}$ and the functional requirement probability distribution $P(f_j \mid Q)$, the model calculates a matching score $g_m$ for each object. The matching score incorporates the causal relationships modeled in $\mathcal{G}$, and the object with the highest score is selected as the final choice. The matching score is computed as:

$$g_m = \sum_{j=1}^{K} P(f_j \mid Q) \cdot \cos(o_m, f_j), \tag{6}$$

where $\cos(o_m, f_j)$ is the similarity between candidate object nodes $o_m$ and functional requirement nodes $f_j$, which is computed using cosine similarity, and the object with the highest score $o^*$ is selected as the final choice. By explicitly constructing the causal graph, the model accurately captures the causal relationships between functional requirements and object features, while removing irrelevant paths to enhance object selection accuracy and efficiency.

### 3.3 CAUSAL-VISUAL FEATURE ALIGNMENT

After obtaining the causal graph $\mathcal{G}$, we further integrate causal features with the geometric–semantic features of the 3D scene point cloud to achieve collaborative reasoning between functional inference and visual perception. Specifically, we first employ a 3D point cloud encoder (e.g., PointNet++) to extract the scene representation $V \in \mathbb{R}^{m \times d}$, where $m$ denotes the number of candidate objects in the point cloud and $d$ is the feature dimension. Meanwhile, based on the functional requirements extracted in the Intent Causal Parsing (represented by the probability distribution $P(f_j \mid Q)$), we obtain a set of causal features $C \in \mathbb{R}^{K \times d}$, where $K$ corresponds to the number of functional requirements. These causal features $C$ are obtained as follows:

$$C_j = P(f_j \mid Q) \cdot \text{Embed}(f_j), \quad j = 1, 2, \ldots, K, \tag{7}$$

where $\text{Embed}(f_j)$ represents the embedding of the $j$-th functional requirement $f_j$ into a $d$-dimensional vector space, and $P(f_j \mid Q)$ is the probability distribution obtained from the Intent Causal Parsing. The resulting set of causal features $C$ is then used for the subsequent fusion with the 3D point cloud features. To align functional requirements with candidate objects, we adopt a cross-attention:

$$\tilde{V} = \text{CrossAttn}(V, C), \tag{8}$$

where $\text{CrossAttn}(\cdot)$ takes the causal features $C$ as queries to dynamically update the visual features $V$, resulting in fused causal-visual representations $\tilde{V}$. On this basis, we introduce a bidirectional consistency constraint to enable mutual verification between causal reasoning and visual reasoning:

$$\mathcal{L}_{\text{cons}} = \sum_{m=1}^{M} \left( \frac{1}{K} \sum_{j=1}^{K} \cos(\tilde{v}_m, C_j) + \frac{1}{K} \sum_{j=1}^{K} \cos(\tilde{v}_m, f_j) \right), \tag{9}$$

where $\tilde{v}_m$ denotes the updated visual feature, $C_j$ represents the causal feature of the $j$-th functional requirement, and $f_j$ is the representation of the functional requirement. Here, $\cos(\cdot, \cdot)$ refers to the cosine similarity. This constraint enforces bidirectional alignment between visual features, causal reasoning, and functional requirements, thereby enhancing the overall accuracy and robustness of the reasoning process. Unlike directly computing matching scores, the fused representations are further utilized to iteratively update the causal graph $\mathcal{G}$, dynamically adjusting node embeddings and edge weights based on the consistency between visual and causal information. Through this iterative updating process, the causal graph evolves alongside the fused representations, reinforcing reliable associations between functional requirements and candidate objects, and enhancing the robustness of subsequent object selection. More reasoning details can be found in Appendix B.3.

Table 1: 3D Intention Grounding results on Intent3D's val set. [0] indicates the zero-shot results. The best results are in **bold**, and the second-best results are underlined.

| Method | Detector | Top1-Acc@0.25 | Top1-Acc@0.5 | AP@0.25 | AP@0.5 |
|---|---|---|---|---|---|
| BUTD-DETR Jain et al. (2022) | GroupFree Liu et al. (2021) | 47.12 | 24.56 | 31.05 | 13.05 |
| EDA Wu et al. (2023) | GroupFree Liu et al. (2021) | 43.11 | 18.91 | 14.02 | 5.00 |
| 3D-ViSTA Zhu et al. (2023) | Mask3D Schult et al. (2022) | 42.76 | 30.37 | 36.1 | 19.93 |
| Chat-3D-v2[0] Huang et al. (2023) | PointGroup Jiang et al. (2020) | 5.86 | 5.24 | 0.15 | 0.13 |
| Chat-3D-v2 Huang et al. (2023) | PointGroup Jiang et al. (2020) | 36.71 | 32.78 | 3.23 | 2.58 |
| IntentNet Kang et al. (2024) | GroupFree Liu et al. (2021) | 58.34 | 40.83 | 41.90 | 25.36 |
| Ours | GroupFree Liu et al. (2021) | **60.71** | **43.24** | **42.65** | **27.33** |

## 4 EXPERIMENTS

In the experiments, for the 3D Intention Grounding (3D-IG) task, we follow the settings of Intent3D Kang et al. (2024) to evaluate the model's cross-modal reasoning ability between natural language intention understanding and 3D object grounding. We conducted comparative experiments on both the validation and test sets and performed fair comparisons with multiple baseline models. Additionally, we conducted further evaluations on 3D Visual Grounding task to comprehensively assess the proposed method's effectiveness in terms of accuracy and robustness.

### 4.1 EXPERIMENTAL SETUP

**Dataset.** In our experiments, for the 3D Intention Grounding (3D-IG) task, we adopt the official evaluation setting of the Intent3D dataset Kang et al. (2024). Intent3D is primarily derived from the ScanNet Dai et al. (2017) dataset and contains 44,990 intention texts across 1,042 ScanNet scenes. The dataset covers large-scale indoor 3D environments paired with natural language intention descriptions, providing an effective benchmark for assessing a model's cross-modal reasoning ability between intention understanding and 3D object grounding. Additionally, we validate the effectiveness of our method on the Nr3D/Sr3D datasets for 3D Visual Grounding task, with more detailed descriptions of the datasets provided in the Appendix B.4.1. arison of Training Time, Infere **Evaluation Metric.** Consistent with baseline methods, we evaluate our model using Top-1 Accuracy and Average Precision (AP). Top-1 Accuracy measures the correctness of the model's highest-confidence prediction, while AP assesses detection performance across varying confidence thresholds. In experiments, we compute the Intersection over Union (IoU) between predicted and ground-truth bounding boxes, and a prediction is considered correct if the IoU exceeds 0.25 or 0.5.

### 4.2 IMPLEMENTATION DETAILS

To ensure a fair comparison, we adopt the same point cloud feature extractor, PointNet++ Qi et al. (2017), and the text feature extractor, RoBERTa Liu et al. (2019), as IntentNet. The model is trained for a total of 120 epochs on four NVIDIA 3090 GPUs with 24 GB of memory, using a batch size of 16, and the overall training time is approximately 40 hours. In terms of input settings, the number of point cloud tokens is fixed at 1024, and the maximum length of text tokens is set to 256. During training, our method parses intention causal chains via Intent Causal Parsing and builds a causal graph through Intent Causal Grounding to align functional requirements with object features, enabling causal reasoning. More implementation details are provided in the Appendix B.4.2.

### 4.3 COMPARISON WITH THE STATE OF THE ART

**Validation on 3D Intention Grounding.** Tables 1 and 2 present a quantitative comparison of our method on the Intent3D validation and test sets. Following the Intent3D evaluation protocol, we compare three categories of approaches: expert models (e.g., BUTD-DETR Jain et al. (2022) and EDA Wu et al. (2023)), foundation models (e.g., 3D-ViSTA Zhu et al. (2023)), and LLM-based models (e.g., Chat-3D-v2 Huang et al. (2023)), as well as the IntentNet method. As shown in Tables 1 and 2, our approach achieves the best performance in both Top-1 Accuracy and Average Precision. For instance, at an IoU threshold of 0.5, our method improves Top-1 Accuracy on the validation set by 6% compared to IntentNet. Our method has three main advantages: Intent Causal Parsing decomposes complex intents into causally linked functional requirements; Intent Causal Grounding precisely maps these requirements to object attributes while removing redundant paths; Causal-Visual Feature Alignment strengthens causal–visual interaction via cross-attention and bidirectional consistency, reducing cross-modal misalignment. We also compare with a two-stage method directly using large language models for intent analysis and object localization; details are presented in Appendix B.5.2.

Table 2: 3D Intention Grounding results on Intent3D's test set. [0] indicates the zero-shot results. The best results are in **bold**, and the second-best results are underlined.

| Method | Detector | Top1-Acc@0.25 | Top1-Acc@0.5 | AP@0.25 | AP@0.5 |
|---|---|---|---|---|---|
| BUTD-DETR Jain et al. (2022) | GroupFree Liu et al. (2021) | 47.86 | 25.74 | 31.41 | 13.46 |
| EDA Wu et al. (2023) | GroupFree Liu et al. (2021) | 44.00 | 19.62 | 14.56 | 5.18 |
| 3D-ViSTA Zhu et al. (2023) | Mask3D Schult et al. (2022) | 43.88 | 31.44 | 37.29 | 22.00 |
| Chat-3D-v2[0] Huang et al. (2023) | PointGroup Jiang et al. (2020) | 5.63 | 4.93 | 0.14 | 0.11 |
| Chat-3D-v2 Huang et al. (2023) | PointGroup Jiang et al. (2020) | 33.46 | 29.32 | 2.67 | 2.10 |
| IntentNet Kang et al. (2024) | GroupFree Liu et al. (2021) | 58.92 | 42.28 | 44.01 | 27.60 |
| Ours | GroupFree Liu et al. (2021) | **61.37** | **45.91** | **45.73** | **30.28** |

Figure 3: **Visualization comparison between our method on the Intent3D datasets.** As shown in the figure, our method on the Intent3D dataset demonstrates superior accuracy.

**Visualization Analysis.** Figure 3 presents a visual comparison between our method and IntentNet. The results clearly demonstrate that our approach achieves higher accuracy in both intent parsing and target object localization. For instance, in the first row of Figure 3, our method successfully identifies and localizes all the sinks in the scene. In contrast, IntentNet only detects one, highlighting our model's superior capability in comprehensive intent understanding. Furthermore, as shown in the second row, our method precisely localizes the target object, the TV stand, while effectively excluding other functionally similar objects, such as the table. This advantage stems from our accurate intent parsing and functional matching, which together ensure precise final localization. Further visual and failure case analyses are provided in Appendix B.5.3 and B.5.4.

**Validation on 3D Visual Grounding. Nr3D and Sr3D Datasets:** To further validate the effectiveness of our method, we evaluated it on the 3D Visual Grounding task. In this setting, Intent Causal Parsing primarily performs step-by-step text decomposition without requiring additional reasoning. In contrast, Intent Causal Grounding and Causal–Visual Feature Alignment enable the model to fully capture object attributes through the construction of a causal graph while further aligning visual and textual features, thereby enhancing performance on the

Table 3: Performance Comparison on 3D Visual Grounding for Nr3D and Sr3D Datasets.

| Methods | Nr3D | | Sr3D | |
|---|---|---|---|---|
| | Overall | Hard | Overall | Hard |
| ReferIt3D Achlioptas et al. (2020) | 35.6 | 27.9 | 40.8 | 31.5 |
| TGNN Huang et al. (2021) | 37.3 | 30.6 | 45.0 | 36.9 |
| 3D-SPS Luo et al. (2022) | 51.5 | 45.1 | 62.6 | 65.4 |
| EDA Wu et al. (2023) | 52.1 | 46.1 | 68.1 | 62.9 |
| MVT Huang et al. (2022) | 55.4 | 49.1 | 64.5 | 58.8 |
| G³-LQ Wang et al. (2024) | 58.4 | 50.7 | 73.1 | 66.3 |
| MiKASA Chang et al. (2024) | 64.4 | 59.4 | 75.2 | 67.3 |
| GPS Jia et al. (2024) | 64.9 | 57.8 | 77.5 | 71.6 |
| MA²TransVG Xu et al. (2024) | 65.2 | 57.6 | 73.9 | 69.3 |
| PQ3D Zhu et al. (2024) | 66.7 | 58.7 | 79.7 | 72.8 |
| Ours | **67.1** | **59.8** | **81.6** | **73.7** |

3D Visual Grounding task. As shown in Table 3, our method achieves the best results both in overall accuracy and on more challenging samples. The performance improvement benefits from the explicit modeling of causal relationships: by understanding functional attributes and their correspondence with visual features, our method can effectively distinguish visually similar objects, enabling more precise localization in complex scenes. To further validate our method's effectiveness in the 3D Visual Grounding task, we conducted ablation experiments in Appendix B.5.1, with detailed analysis provided. **ScanRefer Dataset:** Table 4 presents the detection performance of our method on the ScanRefer dataset. Our approach maintains high accuracy in both the Unique and Multiple settings, demonstrating that the explicit modeling of causal relationships enables the model to fully capture object attributes, understand functional properties, and establish their correspondence with visual features. This allows our method to achieve more precise localization in complex scenes.

Table 4: Performance Comparison on 3D Visual Grounding for ScanRefer DataSet.

| Method | Venue | Unique (~19%) | | Multiple (~81%) | | Overall | |
|---|---|---|---|---|---|---|---|
| | | 0.25 | 0.5 | 0.25 | 0.5 | 0.25 | 0.5 |
| ScanRefer Chen et al. (2020) | ECCV20 | 67.6 | 46.2 | 32.1 | 21.3 | 40.0 | 26.1 |
| EDA Wu et al. (2023) | CVPR23 | 85.8 | 68.6 | 49.1 | 37.6 | 54.6 | 42.3 |
| VPP-Net Shi et al. (2024a) | CVPR24 | 86.1 | 67.1 | 50.3 | 39.0 | 55.7 | 43.3 |
| M3DRef-CLIP Zhang et al. (2023) | ICCV23 | - | 77.2 | - | 36.8 | - | 44.7 |
| G3-LQ Wang et al. (2024) | CVPR24 | 88.6 | 73.3 | 50.2 | 39.7 | 56.0 | 44.7 |
| MCLN Qian et al. (2024b) | ECCV24 | 86.9 | 72.7 | 52.0 | 40.8 | 57.2 | 45.5 |
| MA2TransVG Xu et al. (2024) | CVPR24 | 86.3 | 74.1 | 53.8 | 41.4 | 57.9 | 45.7 |
| ConcreteNet Unal et al. (2024) | ECCV24 | 86.4 | 82.1 | 42.4 | 38.4 | 50.6 | 46.5 |
| D-LISA Zhang et al. (2024) | NeurIPS24 | - | 75.5 | - | 40.0 | - | 46.9 |
| GPS Jia et al. (2024) | ECCV24 | - | 77.9 | - | 42.7 | - | 48.1 |
| Chat-Scene Huang et al. (2024a) | NeurIPS24 | 89.6 | **82.5** | 47.8 | 42.9 | 55.5 | 50.2 |
| PQ3D Zhu et al. (2024) | ECCV24 | 86.7 | 78.3 | 51.5 | 46.2 | 57.0 | 51.2 |
| Robin3D Kang et al. (2025) | ICCV25 | - | - | - | - | **60.8** | **55.1** |
| Ours | - | **90.2** | 78.6 | **53.7** | **46.9** | 59.3 | 52.7 |

## 4.4 ABLATION STUDY

**Component Analysis.** We conducted ablation experiments on the Intent3D test set to evaluate the effectiveness of the three core modules in our framework, summarized in Table 5: Intent Causal Parsing (ICP), Intent Causal Grounding (ICG), and Causal–Visual Feature Alignment (CVFA). The first row reports the baseline performance. The second row shows results after incorporating ICP, where the model can decompose complex natural language in-

Table 5: Ablation analysis of our proposed model. ICP stands for Intent Causal Parsing, ICG represents Intent Causal Grounding, and CVFA denotes Causal-Visual Feature Alignment.

| Methods | | | Metrics | | | |
|---|---|---|---|---|---|---|
| ICP | ICG | CVFA | Top1-Acc@0.25 | Top1-Acc@0.5 | AP@0.25 | AP@0.5 |
| | | | 55.32 | 38.70 | 40.72 | 24.40 |
| ✓ | | | 57.82 | 40.31 | 42.85 | 25.70 |
| | ✓ | | 58.68 | 41.07 | 43.76 | 25.37 |
| | ✓ | ✓ | 59.28 | 42.46 | 43.71 | 27.01 |
| ✓ | ✓ | | 60.99 | 43.67 | 44.25 | 28.38 |
| ✓ | ✓ | ✓ | **61.37** | **45.91** | **45.73** | **30.28** |

tentions into hierarchical functional requirements and directly associate them with objects, achieving a noticeable improvement over the baseline even without causal graph modeling. The third row corresponds to the setup without ICP, where a text encoder is used to extract core verbs and objects for causal graph construction. While this approach yields a slight improvement over the baseline, the gains are limited, as relying solely on verbs and objects introduces logical jumps that reduce overall reasoning accuracy. The fourth row adds CVFA on top of this setup, leading to further improvement and validating the necessity of causal–visual feature alignment. The fifth row presents results with both ICP and ICG, showing significant gains over the second and third rows, highlighting the critical role and synergy of stepwise ICP and explicit causal graph modeling in enhancing performance. Finally, the last row reports the complete model with CVFA, further demonstrating the importance of this module in improving accuracy and robustness. In summary, Table 5 illustrates both the individual effectiveness of each module and the performance benefits arising from their collaborative integration.

Table 6: Comparison of Training Time, Inference Time, Max Memory, and Params (M).

| Method | Training Time (hours) | Inference (ms) | Memory(GB) | Params (M) |
|---|---|---|---|---|
| IntentNet Kang et al. (2024) | 32 | 583 | 7.06 | 148.6 |
| Baseline | 26 | 463 | 6.23 | 120.2 |
| Baseline + ICP | 34 | 623 | 7.34 | 180.6 |
| Baseline + ICP + ICG | 38 | 641 | 7.58 | 182.1 |
| Full Model (Ours) | 40 | 647 | 7.67 | 184.8 |

**Efficiency Analysis.** Table 6 reports the comparison of training time, inference time, memory usage, and model parameters. Training Time refers to the total time required to finish training on four 24GB RTX 3090 GPUs with a batch size of 16. Inference (ms) indicates the average inference latency per batch on a single 24GB RTX 3090 GPU, also with batch size set to 16. Memory denotes the peak GPU memory consumption during inference, while Params represents the total number of trainable parameters. Overall, the Baseline model achieves the best efficiency in both speed and memory usage. As the ICP and ICG modules are progressively added, inference time, memory consumption, and parameter count increase due to the additional causal parsing and alignment computations. Specifically, incorporating the T5-small model in ICP leads to a noticeable increase in inference cost, while the addition of the other two modules has a relatively minor impact on inference speed.

Nonetheless, the model can still be efficiently deployed on a single 24GB GPU, while significantly enhancing its reasoning capabilities and overall robustness.

**Analysis of the Number of Reasoning Steps $T$ in Intent Causal Parsing.** In Intent Causal Parsing, the number of reasoning steps is a key hyperparameter. Too few steps result in incomplete parsing, while too many add complexity, increase computational cost, and risk error propagation. As shown in Figure 4, the model achieves its best performance on the Intent3D test set with four steps, reaching the highest Top1-Acc0.5 and AP0.5. The model effectively extracts core functional requirements and matches them with target objects. Increasing the reasoning steps beyond six provides minimal benefits, complicates causal graph construction, and harms performance. These findings emphasize the need to optimize the number of steps for balancing parsing depth and efficiency.

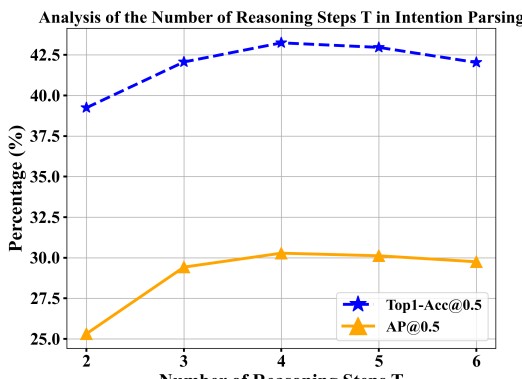

Figure 4: Analysis of the Number of Reasoning Steps $T$ in Intent Causal Parsing.

Table 7: Ablation study on the causal pruning threshold $\theta$.

| Pruning Threshold ($\theta$) | 0.1 | 0.2 | 0.3 | 0.4 | 0.5 |
|---|---|---|---|---|---|
| AP@0.5 | 25.41 | **27.33** | 25.82 | 22.35 | 21.92 |
| Top1-Acc@0.5 | 41.75 | **43.24** | 42.63 | 40.89 | 38.41 |

**Analysis of the Causal Pruning Threshold $\theta$.** To evaluate the effectiveness of our causal pruning strategy and find the optimal threshold, we conducted an ablation study on the pruning threshold $\theta$. As shown in the Table 7, both overly low and overly high thresholds have a negative impact on performance. When $\theta$ is too small (e.g., 0.1), the causal graph retains excessive redundant paths, and the introduced noise interferes with accurate reasoning. Conversely, an overly high threshold (e.g., 0.4, 0.5) over-prunes the graph, removing effective causal links that are crucial for matching key functional attributes. Our model achieved its peak performance (AP@0.5) at $\theta = 0.2$. This indicates that this threshold strikes the best balance between removing irrelevant information and preserving necessary causal connections. This finding validates the importance of our pruning mechanism, as it optimizes the causal graph structure, thereby enhancing the model's accuracy and efficiency.

**Analysis of Intent Parsing and Grounding Accuracy.** To assess the effectiveness of our Intent Causal Parsing (ICP) and Intent Causal Grounding (ICG) modules, we conducted an evaluation on a sampled set of 1,000 intent texts, measuring the Top-1 and Top-5 accuracy of grounding parsed intents to the correct objects. As shown in Table

Table 8: Evaluation of Intent Parsing and Grounding Accuracy.

| Method | Top-1 Acc (%) | Top-5 Acc (%) |
|---|---|---|
| GPT-5 OpenAI (2025) | 73.2 | 87.4 |
| IntentNet Kang et al. (2024) | 76.8 | 88.9 |
| Ours | **83.2** | **91.3** |

8, our method outperforms both GPT-5 OpenAI (2025) and IntentNet Kang et al. (2024), achieving 83.2% Top-1 accuracy and 91.3% Top-5 accuracy. The results indicate that the step-wise decomposition of intents into functional requirements, coupled with explicit causal mapping to objects, significantly improves the model's ability to correctly interpret and ground abstract intentions. Compared to IntentNet, which directly maps intent to objects without explicit causal reasoning, our method demonstrates superior precision in identifying the correct target object, validating the effectiveness of the causal parsing and grounding strategy.

## 5 CONCLUSION

For the 3D Intention Grounding (3D-IG) task, we propose a new method, i.e., Chain-of-Causal Reasoning, which performs intent parsing and grounding along the Chain-of-Causal. Specifically, the method incrementally decomposes complex intentions into functional requirements along the causal chain, clarifying priorities and latent needs, thus forming a causal chain from abstract intentions to object attributes and improving the accuracy of intent understanding. Based on this, we build a causal graph linking functional requirements to object attributes and employ a causal-visual fusion mechanism for deep cross-modal alignment and bidirectional verification. Extensive experiments on 3D Intention Grounding and 3D Visual Grounding tasks demonstrate that our method effectively enhances intent understanding and improves object localization accuracy.

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

# A ACKNOWLEDGEMENT

## A.1 ETHICS STATEMENT

This work adheres to the ICLR Code of Ethics. In this study, no human subjects or animal experimentation were involved. All datasets used were sourced in compliance with relevant usage guidelines, ensuring no violation of privacy. We have taken care to avoid any biases or discriminatory outcomes in our research process. No personally identifiable information was used, and no experiments were conducted that could raise privacy or security concerns. We are committed to maintaining transparency and integrity throughout the research process.

## A.2 REPRODUCIBILITY STATEMENT

We have made every effort to ensure the reproducibility of the results presented in this paper. The experimental setup, including training procedures, model configurations, and hardware details, is described thoroughly in the manuscript. Additionally, we provide a complete account of all experimental details and will release our code along with comprehensive training instructions to facilitate replication.

Furthermore, all datasets used in this study are publicly available, ensuring consistent and reproducible evaluation results.

We believe that these measures will enable other researchers to replicate our work and contribute to further advancements in the field.

## A.3 LLM USAGE

Large Language Models (LLMs) were used to aid in the writing and polishing of the manuscript. Specifically, we used an LLM to assist in refining the language, improving readability, and ensuring clarity in various sections of the paper. The model helped with tasks such as sentence rephrasing, grammar checking, and enhancing the overall flow of the text.

It is important to note that the LLM was not involved in the ideation, research methodology, or experimental design. All research concepts, ideas, and analyses were developed and conducted by the authors. The contributions of the LLM were solely focused on improving the linguistic quality of the paper, with no involvement in the scientific content or data analysis.

# B APPENDIX

## B.1 RELATED WORKS

### B.1.1 CAUSAL LEARNING

Causal relationships have recently been increasingly applied in machine learning and computer vision tasks, as their incorporation enables the development of models that are both more learnable and interpretable. Traditional machine learning relies on modeling statistical correlations in data; however, correlation does not imply causation, which can lead to model failures in scenarios involving interventions or counterfactual reasoning. Causal machine learning addresses this by embedding causal logic, such as causal graphs or potential outcomes frameworks, into data-driven models, providing interpretable and actionable decision support for complex high-dimensional systems. In the 3D point cloud domain, CausalPC Huang et al. (2024b) proposes an adversarial point cloud purification method to enhance model robustness under various attacks. Liu et al. Liu et al. (2022) introduce a method for 3D object reconstruction that constrains the latent space to capture topological causal ordering of underlying factors, thereby simplifying and optimizing subsequent reconstruction steps. In contrast, our approach not only emphasizes causal relationships between intent texts but also captures causal dependencies among visual features. By modeling these multiple interacting causal relationships, our method enhances target recognition and understanding.

## B.2 More Details about Intent Causal Parsing

In Intent Causal Parsing (ICP), we decompose natural language intentions into functional requirements using a prompt template. The template is as follows: 'Given a natural language intention, generate a step-by-step reasoning sequence to identify the functional requirements of the object. The reasoning process should first identify the core demands, then infer the physical attributes, and finally synthesize a complete set of requirements.' For example, as shown in Figure 5, given the intention $Q =$ 'I want something that can support my back to relieve pressure', the model first identifies the core demands (Identify Core Demands), producing $s_1 = \{\text{support back}, \text{relieve pressure}\}$. It then extracts the corresponding object attributes and features for each core demand (Extract Entities and Attributes), resulting in $s_2 = \{\text{object should provide adequate support}\}$ and $s_3 = \{\text{object should be soft}\}$. By integrating these intermediate sequences, the model generates the final reasoning sequence $s_4 = \{\text{object includes both supportiveness and softness}\}$. Finally, the reasoning from the initial intention $s_0 = Q$ to the final sequence $s_4$ is mapped to the complete set of functional requirement nodes. Each node $f_k \in \{f_1, \ldots, f_K\}$ represents a distinct functional property of the object. In this example, the reasoning sequence maps to functional nodes like softness, elasticity, and supportiveness, forming the basis for causal graph construction.

**Natural Language Intention:**
    $Q$ = "I want something that can support my back to relieve pressure"
**Step 1: Identify Core Demands:**
    $s_1 = \{support\ back, relieve\ pressure\}$
**Step 2 : Extract Object Attributes and Features**
For '*support back*':
    $s_2 = \{object\ should\ provide\ adequate\ support\}$
For '*maintain hygiene after touching dirty surfaces*':
    $s_3 = \{object\ should\ be\ soft\}$
**Step 3: Integrate Intermediate Sequences:**
    $s_4 = \{object\ includes\ both\ supportiveness\ and\ softness\}$
**Step 4: Map to Functional Requirement Nodes:**
The final reasoning sequence corresponds to functional nodes such as:
    $f_k \in \{Supportiveness, Softness, Elasticity\}$

**Natural Language Intention:**
    $Q$ = "I want to wash my hands after touching dirty surfaces"
**Step 1: Identify Core Demands:**
    $s_1 = \{wash\ hands, maintain\ hygiene\ after\ touching\ dirty\ surfaces\}$
**Step 2 : Extract Object Attributes and Features**
For '*wash hands*':
    $s_2 = \{object\ should\ provide\ access\ to\ running\ water\ or\ sanitizer\}$
For '*maintain hygiene after touching dirty surfaces*':
    $s_3 = \{object\ should\ enable\ thorough\ cleaning\ and\ have\ antibacterial\ effect\}$
**Step 3: Integrate Intermediate Sequences:**
    $s_4 = \{object\ enables\ proper\ hand\ cleaning\ and\ ensures\ hygiene\}$
**Step 4: Map to Functional Requirement Nodes:**
The final reasoning sequence corresponds to functional nodes such as:
    $f_k \in \{Cleanliness, Antibacterial, Hygiene\}$

Figure 5: Step-wise Intent Causal Parsing Example.

## B.3 More Details about Causal-Graph-Based Reasoning

Our method performs reasoning on the constructed causal graph to achieve the final object selection. This process is not a one-shot operation but rather an iterative refinement loop that integrates linguistic, causal, and visual information to make robust decisions.

First, using the probability distribution $P(f_j \mid Q)$ obtained from the Intent Causal Parsing, the model computes an initial matching score for each candidate object. This preliminary score,

$$g_m = \sum_{j=1}^{K} P(f_j \mid Q) \cdot \cos(o_m, f_j), \tag{10}$$

where $o_m$ and $f_j$ denote the visual feature of the $m$-th candidate object and the feature embedding of the $j$-th functional requirement node, respectively. This score serves as a baseline, establishing a preliminary mapping between functional requirements and object features via cosine similarity.

Next, the model enters an iterative reasoning phase to refine this initial matching. At the core of this phase is the Causal-Visual Feature Alignment (CVFA) module, where causal and visual features are fused. The causal features $C_j$, influenced by the functional requirement probabilities $P(f_j \mid Q)$, serve as queries in a cross-attention mechanism that dynamically updates the visual features of each candidate object, $\tilde{V}$. The updated visual features $\tilde{V}$ now encode information guided by the user's intent. These fused representations are then used to refine the causal graph itself: node embeddings and edge weights are adjusted based on the consistency between the fused visual features and the causal features. For instance, if the fused visual feature of an object strongly aligns with the 'softness' causal feature, the edge linking that object to the 'softness' functional requirement is strengthened. This iterative refinement allows the model to continuously verify its linguistic reasoning against real-world visual evidence. Finally, after refinement, the model computes a final matching score for each candidate object using the updated features:

$$g_m = \text{Match}(\tilde{v}_m, C), \tag{11}$$

where $g_m$ represents the final matching score of the $m$-th candidate object, $\tilde{v}_m$ is its updated visual feature, and $C$ denotes the fused causal feature. Here, $\text{Match}(\cdot, \cdot)$ denotes the similarity computation between the updated visual features and the fused causal features. This score captures not only the initial semantic alignment but also the bidirectional consistency between the visual and causal domains. The object with the highest score is selected as the final output. This reasoning process ensures that the final selection is a robust and verifiable decision grounded in both linguistic and visual information.

### B.4 More details of the experimental settings

#### B.4.1 Dataset

Our main experiments are divided into two parts: the 3D Intention Grounding task and the 3D Visual Grounding task, providing a comprehensive evaluation of the effectiveness of our method.

**3D Intention Grounding (3D-IG) Task:** For this task, we adopt the official evaluation setting of the Intent3D dataset Kang et al. (2024). Intent3D is primarily derived from the ScanNet Dai et al. (2017) dataset and contains 44,990 intention texts across 1,042 ScanNet scenes. On average, each scene contains 61 instances and 43 texts, with each scene-object pair receiving 6 intention texts. The dataset includes 568 distinct verbs and 2,894 distinct nouns. Covering large-scale indoor 3D environments with paired natural language intention descriptions, Intent3D provides an effective benchmark for evaluating a model's cross-modal reasoning ability between intention understanding and 3D object grounding. To ensure fair comparison with existing methods, we strictly follow the official data split, dividing the dataset into a training set (35,850 samples), a validation set (2,285 samples), and a test set (6,855 samples), with no overlapping scenes across splits.

**3D Visual Grounding Task:** We validate our method on the Nr3D and Sr3D datasets, both built upon ScanNet and widely used for 3D visual grounding. Nr3D (Natural Reference in 3D) contains 41,503 human-annotated referring expressions across 707 indoor scenes, where each description uniquely identifies a target object among distractors. Sr3D (Spatial Reference in 3D) contains 83,572 machine-generated expressions based on predefined templates, focusing on spatial relationships between objects. Both datasets provide ground-truth object masks. Performance is evaluated using accuracy, measuring the proportion of correctly localized target objects.

#### B.4.2 Implementation Details

To ensure a fair comparison with existing methods, we adopt a common architecture for the foundational feature extractors, using PointNet++ Qi et al. (2017) for 3D point cloud feature extraction and RoBERTa-base Liu et al. (2019) for natural language intent encoding, consistent with baselines such as IntentNet. Our model is implemented in PyTorch and trained on a distributed system of four NVIDIA 3090 GPUs, each equipped with 24GB of memory, with a total batch size of 16. The intent causal parsing module employs a fine-tuned T5-small model Raffel et al. (2020), with the hidden state dimension of the reasoning step vectors $d_s$ set to 512. We found that setting the number of reasoning steps $T$ in intent causal parsing to 4 yields the best results. The final visual and causal feature dimension $d$ is unified at 256. The entire model is optimized using the AdamW optimizer with a learning rate of $5 \times 10^{-5}$ and a weight decay of $1 \times 10^{-4}$, trained for a total of 120 epochs, taking approximately 40 hours. The total loss is a weighted sum, including grounding loss, causal pruning loss $\mathcal{L}_{\text{pruning}}$, and bidirectional consistency loss $\mathcal{L}_{\text{cons}}$, with the grounding loss weight set to 1.0 and both the causal pruning and bidirectional consistency losses set to 0.1. The causal pruning threshold $\theta$ is set to 0.2. For input settings, the number of point cloud tokens is fixed at 1024, and the maximum length of text tokens is set to 256. During training, our method runs end-to-end, parsing intent causal chains and constructing a causal graph to align functional requirements with object features, thereby achieving a unified causal reasoning process for object selection and localization.

### B.5 More Experiments Analysis

#### B.5.1 Ablation of Components for 3D Visual Grounding Task.

To further assess the effectiveness of our method on the 3D Visual Grounding task, we performed an ablation study to evaluate the individual contributions of the three core modules in our framework,

Intent Causal Parsing (ICP), Intent Causal Grounding (ICG), and Causal–Visual Feature Alignment (CVFA), summarized in Table 9. The first row reports the baseline performance. The second row shows results after incorporating ICP. As the 3D Visual Grounding task does not require explicit reasoning over textual information, ICP only performs step-by-step text decomposition without additional reasoning, resulting in minimal change in overall accuracy compared

Table 9: Ablation analysis of our proposed model on the Nr3D and Sr3D datasets.

| Methods | Nr3D | | Sr3D | |
|---|---|---|---|---|
| | Overall | Hard | Overall | Hard |
| Baseline | 64.7 | 55.1 | 75.3 | 69.0 |
| + ICP | 65.1 | 55.4 | 75.3 | 69.2 |
| + ICG | 66.5 | 58.3 | 81.1 | 71.4 |
| + ICP + ICG | 66.4 | 58.9 | 80.9 | 72.4 |
| + ICP + ICG + CVFA | **67.1** | **59.8** | **81.6** | **73.7** |

to the baseline. The third row presents the performance with ICG, which shows a substantial improvement over the first two rows. This demonstrates the benefit of explicitly modeling the causal relationships from text to objects, enabling the model better to understand functional attributes and their correspondence with visual features, thereby effectively distinguishing visually similar objects and achieving more precise localization in complex scenes. The fourth row reports results with both ICP and ICG, yielding a modest improvement over ICG alone. Finally, the last row presents the performance after adding CVFA, which further enhances the results and confirms the necessity of causal–visual feature alignment. In summary, Table 9 illustrates the individual effectiveness of each module as well as the performance gains from their combined integration, validating that our method is effective not only for 3D Intent Grounding but also for general 3D Visual Grounding tasks.

### B.5.2 ADDITIONAL COMPARISON WITH LARGE LANGUAGE MODELS

To evaluate the effectiveness of our method compared to directly using large language models (LLMs) for intention reasoning, we conducted experiments analogous to those for the IntentNet method, as summarized in Table 10. Specifically, for the LLM-based approach, we adopt a two-stage framework using the same detector as ours to predict bounding boxes and object categories. Given the intention and candidate boxes, the LLM analyzes the intent and selects the most relevant objects. The results are shown in Table 10. As observed, neither GPT nor Gemini achieves satisfactory detection performance. This is primarily due to the complexity of 3D scenes and the hallucination problem of LLMs when decoupled from 3D visual information. Although LLMs like GPT-4 Achiam et al. (2023), GPT-5 OpenAI (2025), and Gemini Team et al. (2023) possess strong textual understanding capabilities, they lack deep integration with 3D visual features. During intent analysis, they may produce "over-reasoning" hallucinations, such as misidentifying semantically related but irrelevant objects as targets. Moreover, they cannot leverage 3D spatial structures, object functionality, or other visual cues to verify their reasoning results. Consequently, the objects selected by LLMs often deviate from the true intent-compliant targets, leading to suboptimal overall detection performance.

Table 10: Compare with the two-stage method based on Large Language Models.

| Method | Detector | Top1-acc@0.25 | Top1-acc@0.5 | Ap@0.25 | Ap@0.5 |
|---|---|---|---|---|---|
| Gemini Team et al. (2023) | GroupFree | 39.21 | 29.34 | 12.17 | 9.36 |
| GPT-4 Achiam et al. (2023) | GroupFree | 41.40 | 28.40 | 15.10 | 7.76 |
| GPT-5 OpenAI (2025) | GroupFree | 42.50 | 29.90 | 14.70 | 8.50 |
| IntentNet Kang et al. (2024) | GroupFree | 58.34 | 40.83 | 41.90 | 25.36 |
| Ours | GroupFree | **60.71** | **43.24** | **42.65** | **27.33** |

### B.5.3 ADDITIONAL VISUALIZATION ANALYSIS.

To further evaluate the effectiveness of our method, we conducted additional visualization experiments, as shown in Figure 6, where the green boxes denote ground-truth objects and the red boxes represent the detections by our model. We present results across various scenes, including bathrooms, living rooms, and bedrooms, which involve diverse functional requirements. We also visualize both single-object and multi-object detection cases. As illustrated in the last row of Figure 6, even when multiple target objects are present, our method can accurately parse the intent and localize the corresponding objects. Furthermore, when the target object is relatively small, for example, the 'toilet paper' in the first column of the third row, our method still achieves precise recognition. These results validate the

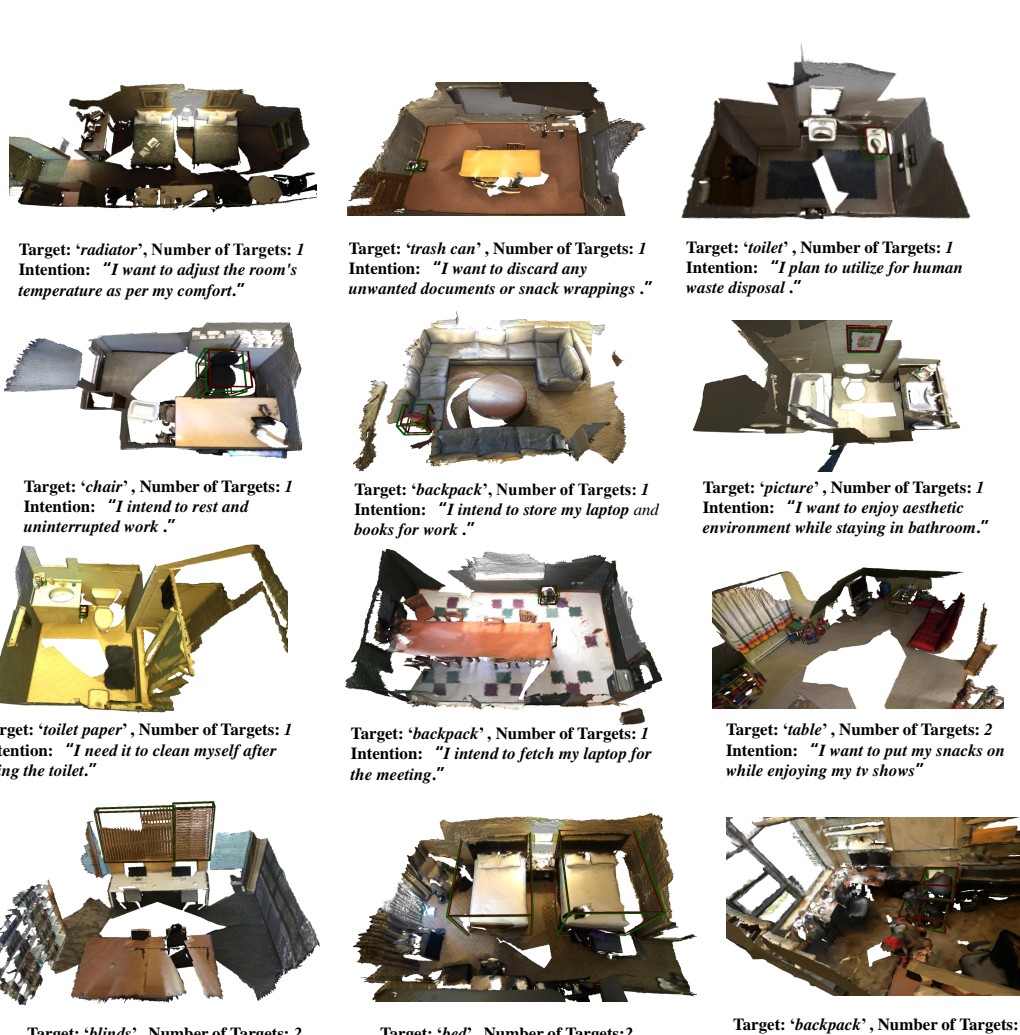

**Target: 'radiator', Number of Targets: 1**
**Intention:** "*I want to adjust the room's temperature as per my comfort.*"

**Target: 'trash can', Number of Targets: 1**
**Intention:** "*I want to discard any unwanted documents or snack wrappings.*"

**Target: 'toilet', Number of Targets: 1**
**Intention:** "*I plan to utilize for human waste disposal.*"

**Target: 'chair', Number of Targets: 1**
**Intention:** "*I intend to rest and uninterrupted work.*"

**Target: 'backpack', Number of Targets: 1**
**Intention:** "*I intend to store my laptop and books for work.*"

**Target: 'picture', Number of Targets: 1**
**Intention:** "*I want to enjoy aesthetic environment while staying in bathroom.*"

**Target: 'toilet paper', Number of Targets: 1**
**Intention:** "*I need it to clean myself after using the toilet.*"

**Target: 'backpack', Number of Targets: 1**
**Intention:** "*I intend to fetch my laptop for the meeting.*"

**Target: 'table', Number of Targets: 2**
**Intention:** "*I want to put my snacks on while enjoying my tv shows*"

**Target: 'blinds', Number of Targets: 2**
**Intention:** "*I intend to adjust the lighting in the room based on my current needs.*"

**Target: 'bed', Number of Targets:2**
**Intention:** "*I plan to get comfortable and sleep at night*"

**Target: 'backpack', Number of Targets: 2**
**Intention:** "*I need it to carry my laptop and office essentials to and from work*"

Figure 6: **More Visualization Results of Our Method on the Intent3D Dataset.** Green boxes indicate ground-truth annotations, while red boxes denote our detection results.

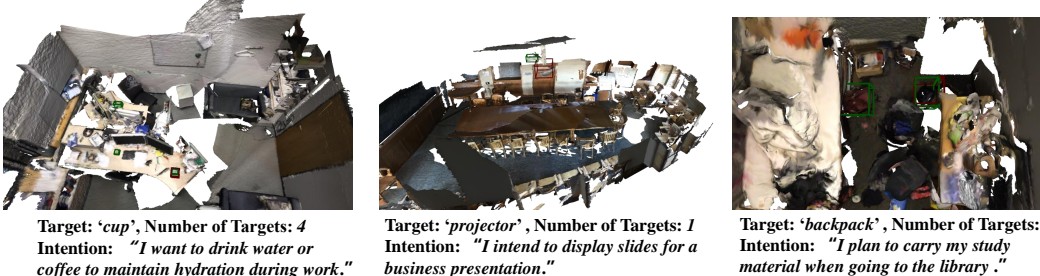

**Target: 'cup', Number of Targets: 4**
**Intention:** "*I want to drink water or coffee to maintain hydration during work.*"

**Target: 'projector', Number of Targets: 1**
**Intention:** "*I intend to display slides for a business presentation.*"

**Target: 'backpack', Number of Targets: 2**
**Intention:** "*I plan to carry my study material when going to the library.*"

Figure 7: **Analysis of Failure Cases on the Intent3D Dataset.** Green boxes indicate ground-truth annotations, while red boxes denote our detection results.

robustness and effectiveness of our approach across diverse and complex scenarios. In summary, the visualization experiments along three dimensions, multi-scene coverage, multi-object adaptation, and small-object recognition, fully demonstrate the effectiveness and robustness of the proposed method in complex 3D intention detection tasks, providing intuitive evidence of its practical applicability.

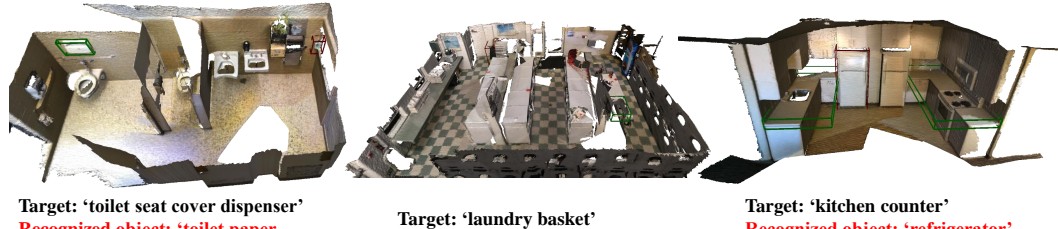

Target: 'toilet seat cover dispenser'
Recognized object: 'toilet paper dispenser'
Intention: 'I need to cover a public toilet seat before use'

Target: 'laundry basket'
Recognized object: 'suitcase'
Intention: 'I want to transport clean clothes back to my room'

Target: 'kitchen counter'
Recognized object: 'refrigerator'
Intention: 'I want to entrepose fresh vegetables to prepare as part of dinner tonight'

Figure 8: **Analysis of Intent Parsing Failure Cases.** Green boxes indicate ground-truth annotations, while red boxes denote our detection results.

### B.5.4 FAILURE CASES AND MODEL LIMITATIONS.

Figure 7 presents several failure cases of our method, highlighting its current limitations. The first type of failure occurs when multiple target objects are present and their sizes are very small. In such cases, the model fails to detect all objects, leading to incomplete localization. This indicates that our method still struggles with densely packed small objects. The second type of failure involves detection box misalignment. In the illustrated scenario, the model parses the intent "I intend to display slides for a business presentation" and identifies the target object as a projector. However, due to inaccuracies in the 3D detection boxes, the final localization is incorrect. This case emphasizes that our method partially relies on the accuracy of candidate boxes: even if the intent parsing is correct, misaligned boxes can result in incorrect object localization. Improving candidate box precision is an important direction for future work. The third type of failure occurs in complex scenes with partial object occlusion, where the model detects only a subset of objects and fails to localize all targets in the scene. This demonstrates that in highly cluttered or occluded environments, our method may not achieve complete object localization.

Figure 8 illustrates several failure cases in intent parsing, highlighting the performance and limitations of our method when the parser makes errors. In the first case, the intent is to cover a toilet seat, and both the toilet paper and the toilet seat cover dispenser share overlapping functionality, specifically the "cover" function, with similar semantics. This causes an error in the intent decomposition, and the model ultimately recognizes the toilet paper dispenser instead of the correct target, the toilet seat cover dispenser. The second case similarly shows a recognition error due to overlapping functionality: the intent is "I want to transport clean clothes back to my room," and during parsing, the model overemphasizes the "transport" function, resulting in the selection of a suitcase rather than the correct laundry basket. In the third case, the intent "I want to entrepose fresh vegetables to prepare as part of dinner tonight" contains the phrase "entrepose fresh vegetables." The model focuses more on the storage function while neglecting the subsequent context "prepare as part of dinner tonight," leading it to recognize the refrigerator used for storing vegetables instead of the kitchen counter used for preparing dinner.

Overall, the failure cases in Figure 7 provide valuable insights for future improvements, such as enhancing the detection of small objects, improving the accuracy of candidate boxes, and strengthening multi-object recognition under occlusion, thereby further boosting the robustness and practical applicability of our method. At the same time, the failure cases in Figure 8 highlight the limitations of our approach during the intent parsing stage. First, when different objects share overlapping functions or have similar semantics, the model is prone to confusion, leading to incorrect function–object matching. Second, the parser may overemphasize certain aspects of complex intents (e.g., "transport" or "storage") while neglecting other critical contextual information, resulting in missed target objects. Finally, these issues are particularly pronounced for intents involving multi-step actions or composite functions, suggesting that future work should focus on improving the iterative reasoning of the causal graph and integrating contextual or multi-modal information to enhance the model's robustness in scenarios with functionally similar intents or complex scenes.

