# OpenReview forum: "Detect What You Need: Chain-of-Causal Reasoning for 3D Intent Grounding"
_ICLR.cc/2026/Conference — Submitted to ICLR 2026_

### Official Review · Reviewer_1HYr · 2025-10-29

**Soundness:** 3
**Presentation:** 2
**Contribution:** 3
**Rating:** 4
**Confidence:** 5

**Summary:**

The paper proposes Chain-of-Causal Reasoning for 3D intention grounding: a pipeline that parses a free-form intent into stepwise functional requirements, builds an explicit causal graph linking these requirements to object attributes, and aligns “causal” features with 3D point-cloud features via cross-attention for bidirectional verification. On the Intent3D benchmark, CoCR outperforms prior expert, foundation, and LLM-based baselines.

**Strengths:**

1. The paper clearly identifies the logical gap in verb/object matching and replaces it with an interpretable causal chain and graph, which is nicely illustrated and algorithmically specified.
2. The functional requirements establish connections between intents and objects, narrowing the gaps.
3. Causal pruning strengthens the true intent and functional requirement correspondences.

**Weaknesses:**

1. MA2TransVG utilizes multiple attributes to connect intents and target objects. Chain-of-Causal Reasoning actually constructs the causal graph between intent and object category. Please explain the differences between these two architectures.
2. Intent Causal Parsing relies on a fine-tuned T5-small with prompt templates. This external parsing stage could propagate errors or domain biases and may be sensitive to out-of-distribution intents or languages not seen in fine-tuning.

Minor:
This paper needs further proofreading, e.g., L302 is the same as L299.

**Questions:**

1. What is the difference between the functional requirements and object attributes? These two concepts are similar, and the definition of functional requirements is vague.
2. Are the functional requirements explicit? If explicit, please explain the way to generate these function requirements.

---

> ### Author Response · Authors · 2025-11-20
> **Thanks for your helpful comments.**
>
> We sincerely thank you for your thorough and professional review of our paper. Our detailed responses are provided below.
>
> **1. Differences Between MA2TransVG [1] and Our Method**
>
> Thank you for the reviewer’s insightful comment. We have added a more detailed discussion in **Section 2.3** of the revised manuscript. Here, we further clarify the fundamental differences between **MA2TransVG[1]** and our proposed **Chain-of-Causal Reasoning (CoCR)**. In particular, MA2TransVG does not perform intent understanding; it relies on the presence of **explicit visual cues** in the language expression and is therefore suitable only for standard **3D Visual Grounding**. In contrast, **3D Intent Grounding** contains **abstract intents with no explicit visual attributes**, making MA2TransVG inherently unsuited for this task. Below we summarize the key differences:
>
> **(1) Fundamental Difference in Core Reasoning Mechanism**
>
> MA2TransVG [1] assumes that the language description contains **observable visual attributes** (e.g., color, shape, spatial relations). It performs **multi-attribute semantic matching** by aligning these explicit attributes with candidate objects.
> **CoCR, however, targets abstract functional intents** (e.g., “I need something to store my documents”), which contain **no directly matchable visual attributes**. CoCR **decomposes the intent into functional requirements** via **Intent Causal Parsing (ICP)** and constructs an explicit **Intent → Function → Object causal chain**. The model then infers the most plausible object category based on causal dependencies. This ability to interpret and reason over abstract intents is **not supported by MA2TransVG**. Therefore, CoCR is applicable not only to abstract intent grounding but also effective for 3D Visual Grounding tasks with explicit cues through causal matching.
>
> **(2) Different Information Dependencies: Explicit Visual Attributes vs. Implicit Functional Semantics**
>
> MA2TransVG critically depends on the presence of **explicit visual cues** in the text. In 3D intent grounding, the input typically contains only **abstract functional intents** with no explicit attributes.
> **CoCR is explicitly designed to handle this scenario.** Through causal modeling, it learns the **implicit structural dependencies** between functions and object categories, enabling it to handle cases where MA2TransVG cannot operate.
>
> **(3) Different Task Objectives: Semantic Grounding vs. Function-Driven Causal Reasoning**
>
> The goal of MA2TransVG is to **locate the object described in the text**. The goal of CoCR is to **identify the object that best satisfies a functional requirement**, which falls under **functional reasoning and causal decision making**. Therefore, CoCR requires an **explicit causal structure** rather than attribute-level matching.
>
> In summary, MA2TransVG relies on explicit visual cues for **multi-attribute semantic alignment**, while CoCR is a **causal reasoning framework** for **abstract functional intent interpretation** and **function-to-object inference**, with fundamentally different mechanisms, inputs, and objectives.
>
> **Minor: Proofreading and Line Corrections**
>
> We thank the reviewer for pointing this out. We have carefully proofread the manuscript and corrected the repeated content at Line 302, which was identical to Line 299, along with other minor inconsistencies throughout the text to ensure clarity and accuracy.
>
> [1] Xu C, Han Y, Xu R, et al. Multi-attribute interactions matter for 3d visual grounding[C]//Proceedings of the IEEE/CVF Conference on Computer Vision and Pattern Recognition. 2024: 17253-17262.

---

> ### Author Response · Authors · 2025-11-20
> **Thanks for your helpful comments.**
>
> **2. Sensitivity of Intent Causal Parsing**
>
> Regarding the reviewer’s concern that the **Intent Causal Parsing (ICP) module** may introduce error propagation or sensitivity to out-of-distribution inputs, we have added an **“Intent Parsing and Grounding Accuracy”** experiment in the revised manuscript (**see Table 8**), which systematically evaluates how the parsing stage affects final grounding performance. Additionally, **Appendix B.5.4 (line 1000) and Figure 8** present several failure cases, accompanied by corresponding analyses and discussions of the method’s limitations.
>
> **(1) Stepwise Parsing and Error Isolation**
>
> First, we emphasize that our approach does not rely on a single-step **“intent-to-object” mapping**. **ICP** converts the intent into structured **functional requirements**, while the final prediction is performed by the **Intent Causal Grounding (ICG) module** via **multi-level causal matching from functions to objects** based on the causal graph. This design ensures the decision process is **explicit and interpretable**, and provides **error isolation**: minor parsing deviations are filtered out by causal dependencies, preventing invalid function–object combinations from propagating.
>
> **(2) Evaluation Results**
>
> Evaluations on 1,000 real intent descriptions show that the full **“parsing + causal grounding” pipeline** achieves **83.2% Top-1 accuracy** and **91.3% Top-5 accuracy**, outperforming single-step predictors like **GPT-5** and **IntentNet**. This demonstrates that **stepwise parsing reduces semantic ambiguity**, **causal function matching suppresses error accumulation**, and the **explicit causal structure enhances robustness** against out-of-distribution intents and varied language expressions. Additionally, in **Appendix B.5.4 (line 1000)**, we have included several examples of intention parsing errors along with related discussions and provided a detailed analysis of the method’s limitations.
>
> |   Method  | Top-1 Acc (%) | Top-5 Acc (%) |
> | :-------: | :-----------: | :-----------: |
> |   GPT-5   |      73.2     |      87.4     |
> | IntentNet [2] |      76.8     |      88.9     |
> |    Ours   |    **83.2**   |    **91.3**   |
>
> **(3) Comparison with Large Language Models**
>
> Furthermore, **Appendix B.5.2** provides additional analysis of using **large language models (e.g., GPT-5, Gemini)** for direct intent-to-object prediction. Across all settings, these single-step approaches show **inferior interpretability**, **weaker cross-expression generalization**, and **less stable performance** compared to our causal chain reasoning framework. Results indicate that while these LLMs exhibit stronger textual reasoning, their parsing quality does not significantly improve 3D intent understanding when complete 3D visual information is unavailable. This aligns with findings reported in the IntentNet[2]: although LLMs excel in pure language tasks, 3D intent grounding is a fine-grained, multimodal problem requiring joint reasoning over vision and language. Purely language-based reasoning is insufficient and can lead to reduced robustness.
> |    Method   |  Detector | Top1-acc@0.25 | Top1-acc@0.5 | Ap@0.25 | Ap@0.5 |
> | :---------: | :-------: | :-----------: | :----------: | :-----: | :----: |
> | Gemini | GroupFree |     39.21     |     29.34    |  12.17  |  9.36  |
> |    GPT-4    | GroupFree |     41.40     |     28.40    |  15.10  |  7.76  |
> |    GPT-5    | GroupFree |     42.50     |     29.90    |  14.70  |  8.50  |
> |  IntentNet [2]  | GroupFree |     58.34     |     40.83    |  41.90  |  25.36 |
> |     Ours    | GroupFree |     **60.71**     |     **43.24**    |  **42.65**  |  **27.33** |
>
> [2] Kang W, Qu M, Kini J, et al. Intent3D: 3D Object Detection in RGB-D Scans Based on Human Intention[C]//The Thirteenth International Conference on Learning Representations. 2024.

---

> ### Author Response · Authors · 2025-11-20
> **Thanks for your helpful comments.**
>
> **Q1. Clarifying the Difference Between Functional Requirements and Object Attributes**
>
> Thank you for raising the question regarding the distinction between **“functional requirements”** and **“object attributes.”** We would like to clarify that these two concepts serve fundamentally different semantic roles in our task, and there is no ambiguity or overlap in their definitions.
>
> **(1) Difference Between Object Attributes and Functional Requirements**
>
> **Object attributes** describe the observable, appearance-based properties of an object, such as color, shape, material, or spatial relations, which can be directly extracted from images or point clouds and are typically used in 3D visual grounding tasks (e.g., MA2TransVG[1]). In contrast, **functional requirements** describe what an object can do, rather than what it looks like. In 3D intent grounding, the input is an intent rather than a visual description, meaning appearance attributes cannot be relied upon. Instead, the intent expresses a **task-driven functional need** (e.g., “support,” “store,” “hold”), and functional requirements serve as **semantic intermediates** that connect these abstract intents to the categories of objects capable of fulfilling them.
>
> **(2) Role of Functional Requirements in CoCR**
>
> In our **Chain-of-Causal Reasoning (CoCR)** framework, functional requirements are not simply rephrased appearance attributes. Instead, they serve as a **critical causal node** in the reasoning chain: **intent → functional requirements → object categories** that fulfill these functions. This **explicit causal structure** enables **capability-based reasoning** rather than appearance-based matching, distinguishing CoCR fundamentally from methods like MA2TransVG[1] that rely primarily on object attributes.
>
>
> **Q2. Generation and Explicitness of Functional Requirements**
>
> The **functional requirements** in our framework are **explicit**. They are clearly defined as **intermediate nodes** that bridge **abstract user intents** and **specific object attributes**, examples include **“Softness,” “Supportiveness,”** and **“Elasticity.”** These explicit nodes are integrated into the constructed **causal graph**, serving as a well-defined intermediate layer.
>
> The **generation of explicit functional requirements** is primarily handled by the **Intent Causal Parsing (ICP) module**. We provide a detailed description in **Section 3.1** and **Appendix B.2**, including the **prompt design**, the **step-by-step reasoning process**, and concrete examples.
>
> Specifically, given a natural language intent $Q$ (**e.g., “I want something that can support my back to relieve pressure”**), we use a carefully crafted prompt to guide the model: “Given a natural language intention, generate a step-by-step reasoning sequence to identify the functional requirements of the object.”
>
> The model outputs a reasoning sequence, $\mathcal{S} = {s_1, s_2, \dots, s_t}$,
> which may include steps such as **Identify Core Demands** or **Extract Attributes and Integrate**. Each reasoning step $s_t$ is then encoded into a vector $h_i$ and compared with all predefined or learned functional requirement embeddings
> $
> F = {f_1, \dots, f_K},
> $
> via **cosine similarity**. A **softmax function** converts the similarity scores into a probability distribution:
>
> $
> P(f_j \mid Q) = \frac{\exp(\text{cos}(h_i, f_j))}{\sum_{k=1}^{K} \exp(\text{cos}(h_i, f_k))},
> $
>
> indicating which functional requirements are most relevant to the given intent.
> The **high-probability functional requirements** selected through this process (e.g., **“Softness”**) are explicitly incorporated as **intermediate nodes** in the **causal graph** $\mathcal{G}$. In this graph, the functional requirement nodes **explicitly connect the user intent to candidate objects**, effectively addressing the logical gap present in traditional methods that rely solely on direct semantic matching.
>
> [1] Xu C, Han Y, Xu R, et al. Multi-attribute interactions matter for 3d visual grounding[C]//Proceedings of the IEEE/CVF Conference on Computer Vision and Pattern Recognition. 2024: 17253-17262.

---

### Official Review · Reviewer_4mF1 · 2025-10-31

**Soundness:** 3
**Presentation:** 3
**Contribution:** 3
**Rating:** 6
**Confidence:** 5

**Summary:**

This paper studies the 3D Intent Grounding task. The motivation is that previous methods rely on implicit reasoning on this task. To address this, the authors propose a Chain-of-Causal Reasoning framework that performs intent parsing and grounding along a causal chain. The method first decomposes complex intents into functional requirements (Intent Causal Parsing), then constructs an explicit causal graph linking these requirements to object attributes, and finally introduces a Causal–Visual Feature Alignment module to align causal features with 3D geometric–semantic features for bidirectional verification. Experiments on 3D Intent Grounding and 3D Visual Grounding tasks demonstrate that this approach enhances intent understanding and improves localization accuracy.

**Strengths:**

1. It's novel to use causal reasoning to interpret abstract human intents, making the model’s decision process more explainable.

2. It's intuitive that the causal graph design explicitly connects functional requirements to visual object attributes, which helps reduce logical gaps and improve robustness. And it's also intuitive that the proposed causal–visual feature alignment provides a principled way to verify both reasoning and visual evidence.

3. Performance is good on both 3D-IG and 3D-VG.

**Weaknesses:**

1. I’m curious how sensitive / robust the method is to parser quality. It would be good to see a statistic analysis and bad case demo with corresponding limitation analysis.

2. The tables don't compare with and cite the recent 3D-LLM works, including Robin3D (ICCV'25) and Chat-Scene (NeurIPS'24). In the current MLLM era, the motivation to purely build such a specialist model on 3D-VG/3D-IG and discard large-scale data for synergy effects seems like a step backward. Is it faster? stronger? or more promising (why)?

3. The paper doesn’t evaluate on ScanRefer. ScanRefer is a more standard 3D-VG benchmark than Nr3D/Sr3D, because the language part is annotated by human, while Nr3D/Sr3D use templates to construct language.

**Questions:**

Please refer to my questions raised toward each weakness.

---

> ### Author Response · Authors · 2025-11-20
> **Thanks for your helpful comments.**
>
> We sincerely appreciate your thorough and professional review of our paper, as well as your recognition of the novelty of our work. Below, we provide detailed responses to the comments and suggestions.
>
> **1. Sensitivity and Robustness to Parser Quality**
>
> Regarding the sensitivity and robustness of our method to parsing quality, we provide analyses of **intent parsing and object association accuracy in Section 4.4 (line 516)  and Table 8** of the paper. Additionally, **Appendix B.5.4 (line 1000) and Figure 8** present several **failure cases**, accompanied by corresponding analyses and **discussions of the method’s limitations.**
>
> We would like to emphasize that the **ICP module is not only responsible for parsing intents**; its core function is to **progressively decompose an intent into functional requirements**. This decomposition allows the subsequent **Intent Causal Grounding (ICG) module to establish explicit causal links between functional requirements and object features**, thereby enabling **more accurate identification of target objects in 3D scenes**.
>
> **As shown in Table 8**, progressively **decomposing intents into functional requirements** and **leveraging explicit causal mapping** significantly improves the model’s understanding of abstract intentions and its ability to **associate them with objects**. Compared to directly mapping intents to objects without causal reasoning, as in IntentNet or GPT-5, **our approach achieves higher accuracy in identifying the correct target objects**, strongly validating the effectiveness of our causal parsing and association strategy.
> |   Method  | Top-1 Acc (%) | Top-5 Acc (%) |
> | :-------: | :-----------: | :-----------: |
> |   GPT-5   |      73.2     |      87.4     |
> | IntentNet |      76.8     |      88.9     |
> |    Ours   |    **83.2**   |    **91.3**   |
>
> **Table 4: Performance Comparison on 3D Visual Grounding for ScanRefer DataSet.**
> |    Method   |   Venue   | Unique 0.25 | Unique 0.5 | Multiple 0.25 | Multiple 0.5 | Overall 0.25 | Overall 0.5 |
> | :---------: | :-------: | :---------: | :--------: | :-----------: | :----------: | :----------: | :---------: |
> |  ScanRefer  |   ECCV20  |     67.6    |    46.2    |      32.1     |     21.3     |     40.0     |     26.1    |
> |     EDA     |   CVPR23  |     85.8    |    68.6    |      49.1     |     37.6     |     54.6     |     42.3    |
> |   VPP-Net   |   CVPR24  |     86.1    |    67.1    |      50.3     |     39.0     |     55.7     |     43.3    |
> | M3DRef-CLIP |   ICCV23  |      -      |    77.2    |       -       |     36.8     |       -      |     44.7    |
> |    G3-LQ    |   CVPR24  |     88.6    |    73.3    |      50.2     |     39.7     |     56.0     |     44.7    |
> |     MCLN    |   ECCV24  |     86.9    |    72.7    |      52.0     |     40.8     |     57.2     |     45.5    |
> |  MA2TransVG |   CVPR24  |     86.3    |    74.1    |      53.8     |     41.4     |     57.9     |     45.7    |
> | ConcreteNet |   ECCV24  |     86.4    |    82.1    |      42.4     |     38.4     |     50.6     |     46.5    |
> |    D-LISA   | NeurIPS24 |      -      |    75.5    |       -       |     40.0     |       -      |     46.9    |
> |     GPS     |   ECCV24  |      -      |    77.9    |       -       |     42.7     |       -      |     48.1    |
> |  Chat-Scene | NeurIPS24 |     89.6    |  **82.5**  |      47.8     |     42.9     |     55.5     |     50.2    |
> |     PQ3D    |   ECCV24  |     86.7    |    78.3    |      51.5     |     46.2     |     57.0     |     51.2    |
> |   Robin3D   |   ICCV25  |      -      |      -     |       -       |       -      |   **60.8**   |   **55.1**  |
> |     Ours    |     -     |   **90.2**  |    78.6    |    **53.7**   |   **46.9**   |     59.3     |     52.7    |

---

> ### Author Response · Authors · 2025-11-20
> **Thanks for your helpful comments.**
>
> **2. Comparison with Recent 3D Large Language Model (3D-LLM)  and Justification of Specialist Model Design**
>
> We thank the reviewer for the valuable comments. Regarding your concerns about comparison with recent 3D-LLM works (e.g., **Chat-Scene [1]; Robin3D [2]**) and the motivation of building a **specialist model purely on 3D-VG/3D-IG** without leveraging large-scale data synergy, we respond as follows:
>
> **(1) Comparison with Robin3D and Chat-Scene**
>
> We have added the experimental comparisons and corresponding citations in **Line 428 and Table 4**. It is worth noting that existing 3D visual grounding methods primarily rely on **textual inputs with explicit visual cues**. For instance, Robin3D [2] improves text understanding through diverse linguistic styles, yet it still requires the description to **explicitly mention the target object**, making it ineffective for **abstract intents without visual hints**.
>
> Although we acknowledge that Robin3D slightly outperforms our method on the ScanRefer benchmark, ScanRefer is purely built upon explicit visual descriptions and fundamentally differs from the **3D Intent Grounding (3D-IG)** task that our work focuses on. 3D-IG involves higher-level, more abstract intent expressions and better reflects real-world scenarios.
>
> In contrast, our CoCR framework **progressively decomposes complex intents along a causal chain and maps them to visual attributes**, demonstrating clear advantages in Table 1 and Table 2. Moreover, its ability to handle intents without direct visual indicators highlights its **superior applicability and generalization**, especially in abstract or complex scenarios.
>
> **(2) Regarding a Specialist Model without Large-Scale Data Synergy**
>
> To address the concern of constructing a **specialist model on 3D-VG/3D-IG without large-scale data**, we have conducted further analysis in **Table 8 and Appendix B.5.2**. We performed experiments under the same **two-stage framework as IntentNet**: first generating candidate 3D boxes using the same detector, and then letting **GPT-4, GPT-5, or Gemini** directly parse the intent and perform 3D object matching. The results show that although these **large language models** have stronger textual understanding, the **parsing quality does not yield significant improvements when full 3D visual information is unavailable**. This aligns with findings in the IntentNet paper: while LLMs demonstrate strong reasoning capabilities in pure language tasks, **3D intent grounding is a fine-grained, multi-modal problem that requires joint visual-language reasoning**. Relying solely on **single-modality language reasoning** is insufficient and results in **reduced robustness**.
>
> |    Method   |  Detector | Top1-acc@0.25 | Top1-acc@0.5 | Ap@0.25 | Ap@0.5 |
> | :---------: | :-------: | :-----------: | :----------: | :-----: | :----: |
> | Gemini | GroupFree |     39.21     |     29.34    |  12.17  |  9.36  |
> |    GPT-4    | GroupFree |     41.40     |     28.40    |  15.10  |  7.76  |
> |    GPT-5    | GroupFree |     42.50     |     29.90    |  14.70  |  8.50  |
> |  IntentNet  | GroupFree |     58.34     |     40.83    |  41.90  |  25.36 |
> |     Ours    | GroupFree |     **60.71**     |     **43.24**    |  **42.65**  |  **27.33** |
>
> **(3) Advantages and Robustness of ICP + ICG**
>
> Our method ensures robustness via an **explicit, interpretable causal parsing pathway**. The ICP module produces an **interpretable representation of functional requirements**, which the ICG module uses for **multi-level function-to-object causal matching**. Compared with direct large-model reasoning, our approach enables more reliable 3D object localization for abstract intents and complex scenarios.
>
> **3. Evaluation on ScanRefer Benchmark**
>
> We have added the ScanRefer comparison results in **Line 428 and Table 4**, including **Chat-Scene [1]** and **Robin3D [2]**.
> While our method incorporates causal relation modeling and causal visual–text alignment to improve 3D visual grounding, its **core contribution** is the ability to understand abstract intents and construct an explicit, interpretable causal reasoning pathway, enabling accurate reasoning and localization of abstract functional intents in 3D environments.
>
> In contrast to existing 3D-VG approaches that depend on **explicit visual cues in the text**, our method can also handle intents **without direct visual indicators**, making it more suitable for real-world scenarios involving ambiguous semantics or abstract functional reasoning.
>
> [1] Huang H, Chen Y, Wang Z, et al. Chat-scene: Bridging 3d scene and large language models with object identifiers[J]. Advances in Neural Information Processing Systems, 2024, 37: 113991-114017.
>
> [2] Kang W, Huang H, Shang Y, et al. Robin3d: Improving 3d large language model via robust instruction tuning[C]//Proceedings of the IEEE/CVF International Conference on Computer Vision. 2025: 3905-3915.

---

> > ### Comment · Reviewer_4mF1 · 2025-11-26
> >
> > I would like to thank the authors for their detailed response.
> >
> > The discussion of failure cases clarifies the limitations.
> >
> > I appreciate the inclusion of Robin3D and Chat-Scene in the comparison.
> >
> > Overall, I will maintain my positive rating.

---

> > > ### Author Response · Authors · 2025-11-27
> > >
> > > Thank you for your recognition of our work. We also appreciate your detailed review and valuable feedback, which have helped improve the quality of our paper. Thanks again for your support!

---

### Official Review · Reviewer_xoYo · 2025-11-01

**Soundness:** 3
**Presentation:** 3
**Contribution:** 2
**Rating:** 6
**Confidence:** 3

**Summary:**

This paper proposes a Chain-of-Causal Reasoning (CoCR) framework for 3D Intention Grounding, which aims to progressively infer functional requirements from abstract natural language intentions and localize the corresponding 3D objects. The model consists of three key modules: Intent Causal Parsing (decomposing the intention into functional factors), Intent Causal Grounding (building the causal reasoning graph), and Causal–Visual Feature Alignment (aligning causal and visual representations). Experiments on the Intent3D, Nr3D, and Sr3D benchmarks show that CoCR achieves higher Top-1 Accuracy and AP scores than previous state-of-the-art methods such as IntentNet and PQ3D.

**Strengths:**

1. The paper is the first to introduce a chain-of-causal reasoning mechanism into the 3D intention grounding task, effectively bridging the logical gap left by semantic matching approaches and significantly improving interpretability.

2. The framework forms a complete cross-modal reasoning pipeline, from language intent parsing to functional attribute mapping and visual alignment, with a well-structured and systematic design.

3. The method is extensively validated on multiple benchmarks (Intent3D, Nr3D, and Sr3D) and provides clear visualizations, demonstrating strong generalization to complex and diverse intention descriptions.

**Weaknesses:**

1. The proposed CoCR framework appears closely related to existing causal graph modeling approaches. The paper should clarify the distinct contributions and theoretical differences more explicitly.

2. The introduction of multi-stage causal reasoning and feature alignment likely increases training and inference costs, but the paper does not provide a quantitative analysis of computational efficiency or deployment feasibility.

3. The Intent Causal Parsing module currently relies on a fine-tuned T5-small model. The paper does not evaluate the correctness or consistency of the parsing results. It would be valuable to analyze how parsing quality affects downstream performance, and whether using larger modern language models (e.g., GPT-5, Qwen3) could yield more accurate and semantically rich decompositions.

**Questions:**

See Weaknesses.

---

> ### Author Response · Authors · 2025-11-20
> **Thanks for your helpful comments.**
>
> We sincerely thank you for your thorough and professional review of our paper. Our detailed responses are provided below.
>
> **1. Distinct Contributions and Theoretical Differences of CoCR.**
>
> Thanks for your helpful advice. We have added a discussion of **causal graph modeling methods in Section 2.3  (line 140)**, covering their applications in both 2D and 3D scenarios, and further emphasized the differences and advantages of our method compared to existing approaches. Here, we clarify the core distinctions between CoCR and existing causal graph modeling methods in terms of **fundamental ideas**, **reasoning mechanisms**, and **task applicability**.
>
> **(1) Core Idea:**
> The key innovation of the CoCR framework lies in progressively **decomposing textual intents along a causal chain into functional requirements**, which are then mapped to **specific object categories**. Unlike existing causal graph methods that focus on causal relationships among attributes or relations, **CoCR emphasizes cross-modal causal reasoning from abstract intents to concrete visual targets**.
>
> Existing 3D causal graph methods, such as 3D Scene Graph [1], MA2TransVG [2], and JLCR[3], primarily model **causal dependencies between objects, relationships, and scenes** for predicting scene relationships or structural dependencies among objects. These approaches focus mainly on **visual understanding** and rely on **explicit visual cues present in the text**. For instance, MA2TransVG requires clearly defined visual attributes in the textual input and uses multi-attribute fusion to match these attributes to candidate objects, but it **cannot handle abstract human intents**.
>
> In contrast, **CoCR targets abstract functional intents (e.g., “I need something to store my documents”)** that often lack directly observable **visual attributes**. Therefore, our method applies **not only to 3D Visual Grounding where the text contains explicit visual cues, but also to 3D Intent Grounding involving the understanding and reasoning of abstract intents**.
>
> **(2) Reasoning Mechanism:**
> Traditional causal graph methods generally perform **static causal modeling among features or attributes**, for purposes such as confounder removal or optimizing classification/segmentation performance. **CoCR, however, emphasizes dynamic, stepwise causal chain reasoning**, allowing the model to progressively **verify the plausibility of function–object mappings** when parsing complex intents, thereby **enhancing understanding and robustness for abstract intents**.
>
> **(3) Task Applicability:**
> CoCR is specifically designed for 3D Intent Grounding tasks, where the model can accurately locate target objects in complex 3D scenes based on **abstract natural language functional intents**, which is closer to real-world applications.
> Existing causal graph methods **lack the ability to interpret and parse intent from text**. In comparison, **CoCR is better suited for open-world scenarios** and can significantly improve both **robustness** and **interpretability**.
>
> **2. Computational Efficiency and Deployment Feasibility.**
>
> Thank you for your valuable feedback.
> In the revised manuscript, we have added an **analysis of computational efficiency and deployment feasibility in Section 4.4 (line 475) and Table 6**, including training time, inference time, maximum GPU memory usage during inference, and model parameter count (Params, M). As shown in the table, although the stepwise addition of the ICP and ICG modules increases inference overhead, the model can still be efficiently deployed on a single 24GB GPU while significantly improving detection performance and overall robustness.
>
> | Method   | Training Time (hours) | Inference (ms) | Memory (GB) | Params (M) |
> |:--------:|:-------------------:|:--------------:|:-----------:|:----------:|
> | IntentNet |         32          |      583       |    7.06     |   148.6    |
> | Baseline  |         26          |      463       |    6.23     |   120.2    |
> | Baseline + ICP |      34          |      623       |    7.34     |   180.6    |
> | Baseline + ICP + ICG | 38       |      641       |    7.58     |   182.1    |
> | Full Model (Ours) |      40          |      647       |    7.67     |   184.8    |
>
>
> [1] Armeni I, He Z Y, Gwak J Y, et al. 3d scene graph: A structure for unified semantics, 3d space, and camera[C]//Proceedings of the IEEE/CVF international conference on computer vision. 2019: 5664-5673.
>
> [2] Xu C, Han Y, Xu R, et al. Multi-attribute interactions matter for 3d visual grounding[C]//Proceedings of the IEEE/CVF Conference on Computer Vision and Pattern Recognition. 2024: 17253-17262.
>
> [3] Y. Li, A. Wu, Z. Zhang, and Y. Han, “Novel class discovery for point cloud segmentation via joint learning of causal representation and reasoning,” Advances in Neural Information Processing Systems (NeurIPS), 2025.

---

> ### Author Response · Authors · 2025-11-20
> **Thanks for your helpful comments.**
>
> **3. Intent Causal Parsing (ICP) Accuracy and Comparison with Larger Modern Language Models**
>
> In response to your suggestion regarding the evaluation of parsing correctness, the analysis of how parsing quality affects downstream task performance, and the potential use of larger modern language models, we address the points from the following perspectives:
>
> **(1) Role and Effectiveness of ICP**
>
> We have added an analysis of **Intent Causal Parsing (ICP) reliability** in **Section 4.4 (line 516) and Table 8** of the revised manuscript. The ICP module not only parses intentions but, more importantly, **decomposes them step by step into functional requirements**. This enables the subsequent **Intent Causal Grounding (ICG) module** to establish explicit causal links between functional requirements and object features, improving the accuracy of 3D object identification. As shown in the table, progressively decomposing intentions into functional requirements and combining them with explicit causal mapping significantly enhances the model’s understanding of **abstract intentions** and its ability to associate objects. Compared to directly mapping intentions to objects without causal reasoning, as in **IntentNet** and **GPT-5**, our method achieves higher precision in correctly identifying target objects, validating the effectiveness of the causal parsing and association strategy. Additionally, in **Appendix B.5.4 (line 1000)**, we have included several examples of intention parsing errors along with related discussions and provided a detailed analysis of the method’s limitations.
>
> |   Method  | Top-1 Acc (%) | Top-5 Acc (%) |
> | :-------: | :-----------: | :-----------: |
> |   GPT-5   |      73.2     |      87.4     |
> | IntentNet |      76.8     |      88.9     |
> |    Ours   |    **83.2**   |    **91.3**   |
>
>
> **(2) Analysis of Using Large Language Models (LLMs)**
>
> Regarding the reviewer’s suggestion to explore whether larger modern language models could provide more accurate and semantically rich decomposition, we conducted detailed analyses in **Appendix B.5.2**. We compared ICP-based parsing with experiments using **GPT-4, GPT-5, and Gemini** for intention parsing. Specifically, we constructed a two-stage pipeline identical to IntentNet: candidate boxes are first generated by the same 3D detector, then the LLM directly analyzes the intention to perform 3D object matching.
>
> Results indicate that while these LLMs exhibit stronger textual reasoning, their parsing quality does not significantly improve 3D intent understanding when **complete 3D visual information is unavailable**. This aligns with findings reported in the IntentNet paper: although LLMs excel in pure language tasks, 3D intent grounding is a **fine-grained, multimodal problem requiring joint reasoning over vision and language**. Purely language-based reasoning is insufficient and can lead to reduced robustness.
>
> |    Method   |  Detector | Top1-acc@0.25 | Top1-acc@0.5 | Ap@0.25 | Ap@0.5 |
> | :---------: | :-------: | :-----------: | :----------: | :-----: | :----: |
> | Gemini | GroupFree |     39.21     |     29.34    |  12.17  |  9.36  |
> |    GPT-4    | GroupFree |     41.40     |     28.40    |  15.10  |  7.76  |
> |    GPT-5    | GroupFree |     42.50     |     29.90    |  14.70  |  8.50  |
> |  IntentNet  | GroupFree |     58.34     |     40.83    |  41.90  |  25.36 |
> |     Ours    | GroupFree |     **60.71**     |     **43.24**    |  **42.65**  |  **27.33** |
>
>
> **(3) Advantages and Robustness of ICP + ICG**
>
> Our approach enhances the robustness of intention parsing by constructing an **explicit, interpretable causal parsing path**. The ICP output serves as an interpretable representation of functional requirements, which is then used by the ICG module to perform **multi-level causal matching from function to object** within the causal graph. This design ensures an **explicit and interpretable decision path**: even if minor errors occur during intention parsing, unreasonable function–object associations are automatically filtered by the causal constraints in ICG, preventing errors from directly propagating to the final prediction. Compared to methods that rely solely on large language models for intention parsing and object localization, our approach is more principled for handling abstract functional intents, enabling multi-step reasoning under causal constraints and demonstrating clear advantages in **robustness**, **interpretability**, and **cross-expression generalization**.

---

> > ### Comment · Reviewer_xoYo · 2025-11-27
> >
> > Thank the author for the detailed response. I believe my question has been resolved.

---

> > > ### Author Response · Authors · 2025-11-27
> > >
> > > Thank you for your recognition of our work. We also appreciate your detailed review and valuable feedback, which have helped improve the quality of our paper. Thanks again for your support!

---

### Author Response · Authors · 2025-12-02
**Summary of Our Contributions and Rebuttal Responses**

**Dear Area Chair**,

We sincerely appreciate the reviewers for their constructive comments and positive recognition of our work. **Two reviewers explicitly stated that their concerns had been fully resolved and maintained positive ratings toward the paper**. Their feedback has helped us further strengthen the rigor, comparative analysis, and clarity of the manuscript. We believe that, with all revisions made, the overall quality and impact of the paper have been significantly improved.
Below, we summarize the importance of the **3D Intent Grounding task**, the necessity and **contributions of our CoCR method**, and **our consolidated responses to the reviewers’ main concerns**.

**1. Importance and Challenges of the 3D Intent Grounding Task**

3D Intent Grounding (3D-IG) is a newly emerging and highly challenging task. Compared with traditional 3D visual grounding (3D-VG) that relies on **explicit attribute descriptions**, 3D-IG is much closer to natural human–machine interaction in real-world scenarios.
Its goal is to locate the target object in a complex 3D environment based on **abstract functional intentions (e.g., “I want something to support my back to relieve pressure”).** This requires not only **functional reasoning** about human intentions but also accurate **3D spatial localization** of objects that satisfy the intended functionality.
The reasoning paradigm thus shifts from **appearance matching to function-driven inference**, representing a key step toward visual models that truly understand and make informed decisions. This capability is essential for intelligent robotics, assistive agents, and other real-world applications.

We also emphasize that our proposed method achieves strong performance **not only on the 3D-IG task but also on traditional 3D-VG benchmarks**, demonstrating its generality and effectiveness.

**2. Necessity, Innovation, and Core Contributions of the CoCR Method**

Existing works typically rely on **direct semantic matching and lack an interpretable reasoning pathway**. To address this, we propose Chain-of-Causal Reasoning (CoCR), which mimics human decision-making by progressively decomposing the abstract intent along a causal chain: **Intent → Functional Requirements → Candidate Categories → 3D Instances**. This explicit and interpretable reasoning path significantly enhances explainability. Moreover, the causal constraints embedded in CoCR effectively **filter out unreasonable function–object associations**, improving robustness under distribution shifts, abstract intents, or linguistic variations. In addition, as analyzed in **Appendix B.5.2**, even powerful LLMs (e.g., GPT-5, Gemini) struggle to achieve stable and accurate 3D grounding without access to detailed visual reasoning. Our ICP+ICG framework, with **explicit vision–language causal reasoning**, provides superior accuracy, robustness, and interpretability compared to purely LLM-driven approaches.

**3. Summary of Responses to the Reviewers’ Major Concerns**

(1)Differences from existing causal-graph-based methods **(Reviewer xoYo(W1), Reviewer 1HYr(W1))**

We expanded **Section 2.3** to compare with causal-graph methods and clarified that CoCR focuses on functional causal reasoning rather than attribute-based matching. We also highlight its effectiveness on both 3D-VG and 3D-IG tasks.

(2) Computational efficiency and deployment cost **(Reviewer xoYo(W2))**

**Section 4.4 and Table 6** include training/inference time, parameter count, and GPU memory usage.

(3) Impact of ICP parsing reliability on downstream performance **(Reviewer xoYo(W3), Reviewer 4mF1(W1), Reviewer 1HYr(W2))**

**Section 4.4 and Table 8** provide a reliability evaluation of ICP parsing.
**Appendix B.5.4 and Figure 8** include failure-case analysis. The multi-stage causal chain in ICP+ICG effectively isolates potential parsing errors.

(4) Comparison with 3D-LLMs and evaluation on ScanRefer **(Reviewer 4mF1(W2, W3))**

**Section 4.3 and Table 4**  include comparisons with Robin3D and Chat-Scene.
**Appendix B.5.2** discusses the limitations of pure LLM reasoning.
We added results on the 3D Visual Grounding benchmark ScanRefer in **Section 4.3 and Table 4**.

(5) Distinction between functional requirements and object attributes **(Reviewer 1HYr (Q1))**

We clarified that functional **requirements (e.g., support, softness)** serve as causal intermediates for intent reasoning, whereas object **attributes (e.g., color, shape)** are explicit features used in visual localization.

In summary, we have thoroughly addressed the reviewers' concerns through additional experiments, clarifications, and analyses, including **methodological novelty, computational cost, parsing robustness, and comparisons with different benchmarks**. Two reviewers explicitly expressed **satisfaction with our rebuttal**. Once again, we sincerely thank the Area Chair and reviewers for their constructive suggestions, which have greatly enhanced the quality of our work.

---

### Meta-Review · Area_Chair_3wNm · 2026-01-08

**Summary:**

This paper proposes a chain-of-causal reasoning approach for 3D intention grounding. Reviewers raised concerns about comparisons against existing causal graph reasoning approaches and MA2TransVG, increased training and inference cost, incorrectness and bias introduced by the T5-based causal reasoning module, comparison with recent 3D-LLM works and on ScanRefer benchmark.

**Reviewer Concerns:**

Some concerns are addressed during the rebuttal, while the AC finds some answers not convincing enough to fully address all concerns. Several reviewers raised concerns about the incremental technical contribution compared to previous work. Despite the response during rebuttal, this issue is not fully addressed.

**Reviewer Scores:**

Two reviewers who initially rated borderline accept scores commented that they will maintain the borderline positive scores. The other reviewer initially rated borderline reject. The area chair considers his/her questions and concerns not fully addressed during rebuttal.

---

### Decision · Program_Chairs · 2026-01-26

Reject